# Can LLMs Express Their Uncertainty? An Empirical Evaluation of Confidence Elicitation in LLMs

**Miao Xiong**[1]*, **Zhiyuan Hu**[1], **Xinyang Lu**[1], **Yifei Li**[3], **Jie Fu**[2], **Junxian He**[2]†, **Bryan Hooi**[1]†
[1] National University of Singapore    [2] The Hong Kong University of Science and Technology
[3] École Polytechnique Fédérale de Lausanne

## Abstract

Empowering large language models (LLMs) to accurately express confidence in their answers is essential for reliable and trustworthy decision-making. Previous confidence elicitation methods, which primarily rely on *white-box access* to internal model information or model fine-tuning, have become less suitable for LLMs, especially closed-source commercial APIs. This leads to a growing need to explore the untapped area of *black-box* approaches for LLM uncertainty estimation. To better break down the problem, we define a systematic framework with three components: *prompting* strategies for eliciting verbalized confidence, *sampling* methods for generating multiple responses, and *aggregation* techniques for computing consistency. We then benchmark these methods on two key tasks—confidence calibration and failure prediction—across five types of datasets (e.g., commonsense and arithmetic reasoning) and five widely-used LLMs including GPT-4 and LLaMA 2 Chat. Our analysis uncovers several key insights: 1) LLMs, when verbalizing their confidence, tend to be *overconfident*, potentially imitating human patterns of expressing confidence. 2) As model capability scales up, both calibration and failure prediction performance improve, yet still far from ideal performance. 3) Employing our proposed strategies, such as human-inspired prompts, consistency among multiple responses, and better aggregation strategies can help mitigate this overconfidence from various perspectives. 4) Comparisons with white-box methods indicate that while white-box methods perform better, the gap is narrow, e.g., 0.522 to 0.605 in AUROC. Despite these advancements, none of these techniques consistently outperform others, and all investigated methods struggle in challenging tasks, such as those requiring professional knowledge, indicating significant scope for improvement. We believe this study can serve as a strong baseline and provide insights for eliciting confidence in black-box LLMs. The code is publicly available at https://github.com/MiaoXiong2320/llm-uncertainty.

## 1 Introduction

A key aspect of human intelligence lies in our capability to meaningfully *express and communicate our uncertainty* in a variety of ways (Cosmides & Tooby, 1996). Reliable uncertainty estimates are crucial for human-machine collaboration, enabling more rational and informed decision-making (Guo et al., 2017; Tomani & Buettner, 2021). Specifically, accurate confidence estimates of a model can provide valuable insights into the reliability of its responses, facilitating risk assessment and error mitigation (Kuleshov et al., 2018; Kuleshov & Deshpande, 2022), selective generation (Ren et al., 2022), and reducing hallucinations in natural language generation tasks (Xiao & Wang, 2021).

In the existing literature, eliciting confidence from machine learning models has predominantly relied on *white-box access* to internal model information, such as token-likelihoods (Malinin & Gales, 2020; Kadavath et al., 2022) and associated calibration techniques (Jiang et al., 2021), as well as model fine-tuning (Lin et al., 2022). However, with the prevalence of large language models, these

---

*Corresponding to: Miao Xiong (miao.xiong@u.nus.edu).
†Equal advising: bhooi@comp.nus.edu.sg, junxianh@cse.ust.hk

methods are becoming less suitable for several reasons: 1) The rise of closed-source LLMs with commercialized APIs, such as GPT-3.5 (OpenAI, 2021) and GPT-4 (OpenAI, 2023), which only allow textual inputs and outputs, lacking access to token-likelihoods or embeddings; 2) Token-likelihood primarily captures the model's uncertainty about the next token (Kuhn et al., 2023), rather than the semantic probability inherent in textual meanings. For example, in the phrase "Chocolate milk comes from brown cows", every word fits naturally based on its surrounding words, but high individual token likelihoods do not capture the falsity of the overall statement, which requires examining the statement semantically, in terms of its claims; 3) Model fine-tuning demands substantial computational resources, which may be prohibitive for researchers with lower computational resources. Given these constraints, there is a growing need to explore *black-box* approaches for eliciting the confidence of LLMs in their answers, a task we refer to as *confidence elicitation*.

Recognizing this research gap, our study aims to contribute to the existing knowledge from two perspectives: 1) explore *black-box* methods for confidence elicitation, and 2) conduct a comparative analysis to shed light on methods and directions for eliciting more accurate confidence. To achieve this, we define a systematic framework with three components: **prompting** strategies for eliciting verbalized confidence, **sampling** strategies for generating multiple responses, and **aggregation** strategies for computing the consistency. For each component, we devise a suite of methods. By integrating these components, we formulate a set of algorithms tailored for confidence elicitation. A comprehensive overview of the framework is depicted in Figure 1. We then benchmark these methods on two key tasks—confidence calibration and failure prediction—across five types of tasks (Commonsense, Arithmetic, Symbolic, Ethics and Professional Knowledge) and five widely-used LLMs, i.e., GPT-3 (Brown et al., 2020), GPT-3.5 (OpenAI, 2021), GPT-4, Vicuna (Chiang et al., 2023) and LLaMA 2 (Touvron et al., 2023b).

Our investigation yields several observations: 1) LLMs tend to be highly overconfident when verbalizing their confidence, posing potential risks for the safe deployment of LLMs (§5.1). Intriguingly, the verbalized confidence values predominantly fall within the 80% to 100% range and are typically in multiples of 5, similar to how humans talk about confidence. In addition, while scaling model capacity leads to performance improvement, the results remain suboptimal. 2) Prompting strategies, inspired by patterns observed in human dialogues, can mitigate this overconfidence, but the improvement also diminishes as the model capacity scales up (§5.2). Furthermore, while the calibration error (e.g. ECE) can be significantly reduced using suitable prompting strategies, failure prediction still remains a challenge. 3) Our study on sampling and aggregation strategies indicates their effectiveness in improving failure prediction performance (§5.3). 4) A detailed examination of aggregation strategies reveals that they cater to specific performance metrics, i.e., calibration and failure prediction, and can be selected based on desired outcomes (§5.4). 5) Comparisons with white-box methods indicate that while white-box methods perform better, the gap is narrow, e.g., 0.522 to 0.605 in AUROC (§B.1). Despite these insights, it is worth noting that the methods introduced herein still face challenges in failure prediction, especially with tasks demanding specialized knowledge (§6). This emphasizes the ongoing need for further research and development in confidence elicitation for LLMs.

## 2 RELATED WORKS

**Confidence Elicitation in LLMs.** Confidence elicitation is the process of estimating LLM's confidence in their responses without model fine-tuning or accessing internal information. Within this scope, Lin et al. (2022) introduced the concept of verbalized confidence that prompts LLMs to express confidence directly. However, they mainly focus on fine-tuning on specific datasets where the confidence is provided, and its zero-shot verbalized confidence is unexplored. Other approaches, like the external calibrator from Mielke et al. (2022), depend on internal model representations, which are often inaccessible. While Zhou et al. (2023) examines the impact of confidence, it does not provide direct confidence scores to users. Our work aligns most closely with the concurrent study by Tian et al. (2023), which mainly focuses on the use of prompting strategies. Our approach diverges by aiming to explore a broader method space, and propose a comprehensive framework for systematically evaluating various strategies and their integration. We also consider a wider range of models beyond those RLHF-LMs examined in concurrent research, thus broadening the scope of confidence elicitation. Our results reveal persistent challenges across more complex tasks and contribute to a holistic understanding of confidence elicitation. For a more comprehensive discussion of the related works, kindly refer to Appendix C.

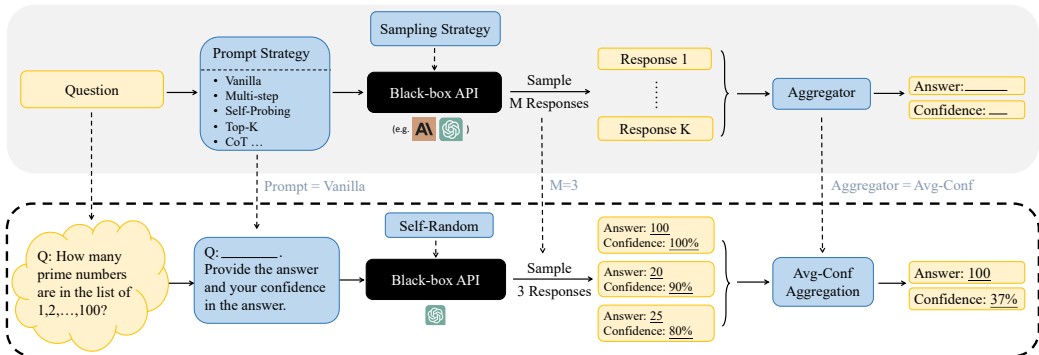

Figure 1: An Overview and example of Confidence Elicitation framework, which consists of three components: prompt, sampling and aggregator. By integrating distinct strategies from each component, we can devise different algorithms, e.g., Top-K (Tian et al., 2023) is formulated using Top-K prompt, self-random sampling with $M = 1$, and Avg-Conf aggregation. Given an input question, we first choose a suitable *prompt* strategy, e.g., the vanilla prompt used here. Next, we determine the number of samples to generate ($M = 3$ here) and *sampling* strategy, and then choose an *aggregator* based on our preference (e.g. focus more on improving calibration or failure prediction) to compute confidences in its potential answers. The highest confident answer is selected as the final output.

## 3 EXPLORING BLACK-BOX FRAMEWORK FOR CONFIDENCE ELICITATION

In our pursuit to explore black-box approaches for eliciting confidence, we investigated a range of methods and discovered that they can be encapsulated within a unified framework. This framework, with its three pivotal components, offers a variety of algorithmic choices that combine to create diverse algorithms with different benefits for confidence elicitation. In our later experimental section (§5), we will analyze our proposed strategies within each component, aiming to shed light on the best practices for eliciting confidence in black-box LLMs.

### 3.1 MOTIVATION OF THE FRAMEWORK

**Prompting strategy.** The key question we aim to answer here is: in a black-box setting, what form of model inputs and outputs lead to the most accurate confidence estimates? This parallels the rich study in eliciting confidences from *human* experts: for example, patients often inquire of doctors about their confidence in the potential success of a surgery. We refer to this goal as verbalized confidence, and inspired by strategies for human elicitation, we design a series of human-inspired prompting strategies to elicit the model's verbalized confidence. We then unify these prompting strategies as a building block of our framework (§3.2). In addition, beyond its simplicity, this approach also offers an extra benefit over model's token-likelihood: the verbalized confidence is intrinsically tied to the semantic meaning of the answer instead of its syntactic or lexical form (Kuhn et al., 2023).

**Sampling and Aggregation.** In addition to the direct insights from model outputs, the variance observed among multiple responses for a given question offers another valuable perspective on model confidence. This line of thought aligns with the principle extensively explored in prior white-box access uncertainty estimation methodologies for classification (Gawlikowski et al., 2021), such as MCDropout (Gal & Ghahramani, 2016) and Deep Ensemble (Lakshminarayanan et al., 2017). The challenges in adapting ensemble-based methods lie in two critical components: 1) the *sampling strategy*, i.e., how to sample multiple responses from the model's answer distribution, and 2) the *aggregation strategy*, i.e., how to aggregate these responses to yield the final answer and its associated confidence. To optimally harness both textual output and response variance, we have integrated them within a unified framework.

### 3.2 PROMPTING STRATEGY

Drawing inspiration from patterns observed in human dialogues, we design a series of human-inspired prompting strategies to tackle challenges, e.g., overconfidence, that are inherent in the vanilla version of verbalized confidence. See Table 1 for an overview of these prompting strategies and Appendix F for complete prompts.

Table 1: Illustration of the prompting strategy (the complete prompt in Appendix F). To help models understand the concept of confidence, we also append the explanation "Note: The confidence indicates how likely you think your answer is true." to every prompt.

| Method | Prompt |
|---|---|
| Vanilla | Read the question, provide your answer, and your confidence in this answer. |
| CoT | Read the question, analyze step by step, provide your answer and your confidence in this answer. |
| Self-Probing | Question: [...] Possible Answer: [...] Q: How likely is the above answer to be correct? Analyze the possible answer, provide your reasoning concisely, and give your confidence in this answer. |
| Multi-Step | Read the question, break down the problem into K steps, think step by step, give your confidence in each step, and then derive your final answer and your confidence in this answer. |
| Top-K | Provide your $K$ best guesses and the probability that each is correct (0% to 100%) for the following question. |

**CoT.** Considering that a better comprehension of a problem can lead to a more accurate understanding of one's certainty, we adopt a reasoning-augmented prompting strategy. In this paper, we use zero-shot Chain-of-Thought, CoT (Kojima et al., 2022) for its proven efficacy in inducing reasoning processes and improving model accuracy across diverse datasets. Alternative strategies such as plan-and-solve (Wang et al., 2023) can also be used.

**Self-Probing.** A common observation of humans is that they often find it easier to identify errors in others' answers than in their own, as they can become fixated on a particular line of thinking, potentially overlooking mistakes. Building on this assumption, we investigate if a model's uncertainty estimation improves when given a question and its answer, then asked, *"How likely is the above answer to be correct"?* The procedure involves generating the answer in one chat session and obtaining its verbalized confidence in another independent chat session.

**Multi-Step.** Our preliminary study shows that LLMs tend to be overconfident when verbalizing their confidence (see Figure 2). To address this, we explore whether dissecting the reasoning process into steps and extracting the confidence of each step can alleviate the overconfidence. The rationale is that understanding each reasoning step's confidence could help the model identify potential inaccuracies and quantify their confidence more accurately. Specifically, for a given question, we prompt models to delineate their reasoning process into individual steps $S_i$ and evaluate their confidence in the correctness of this particular step, denoted as $C_i$. The overall verbalized confidence is then derived by aggregating the confidence of all steps: $C_{\text{multi-step}} = \prod_{i=1}^{n} C_i$, where $n$ represents the total number of reasoning steps.

**Top-K.** Another way to alleviate overconfidence is to realize the existence of multiple possible solutions or answers, which acts as a normalization for the confidence distribution. Motivated by this, Top-K (Tian et al., 2023) prompts LLMs to generate the top $K$ guesses and their corresponding confidence for a given question.

### 3.3 SAMPLING STRATEGY

Several methods can be employed to elicit multiple responses of the same question from the model: 1) **Self-random**, leveraging the model's inherent randomness by *inputting the same prompt multiple times*. The temperature, an adjustable parameter, can be used to calibrate the predicted token distribution, i.e., adjust the diversity of the sampled answers. An alternative choice is to *introduce perturbations in the questions*: 2) **Prompting**, by paraphrasing the questions in different ways to generate multiple responses. 3) **Misleading**, feeding *misleading* cues to the model, e.g.,"I think the answer might be ...". This method draws inspiration from human behaviors: when confident, individuals tend to stick to their initial answers despite contrary suggestions; conversely, when uncertain, they are more likely to waver or adjust their responses based on misleading hints. Building on this observation, we evaluate the model's response to misleading information to gauge its uncertainty. See Table 11 for the complete prompts.

### 3.4 AGGREGATION STRATEGY

**Consistency.** A natural idea of aggregating different answers is to measure the degree of agreement among the candidate outputs and integrate the inherent uncertainty in the model's output.

For any given question and an associated answer $\tilde{Y}$, we sample a set of *candidate answers* $\hat{Y}_i$, where $i \in \{1, ..., M\}$. The agreement between these candidate responses and the original answer then serves as a measure of confidence, computed as follows:

$$C_{\text{consistency}} = \frac{1}{M} \sum_{i=1}^{M} \mathbb{I}\{\hat{Y}_i = \tilde{Y}\}. \tag{1}$$

**Avg-Conf.** The previous aggregation method does not utilize the available information of verbalized confidence. It is worth exploring the potential synergy between these uncertainty indicators, i.e., whether the verbalized confidence and the consistency between answers can complement one another. For any question and an associated answer $\tilde{Y}$, we sample a candidate set $\{\hat{Y}_1, ...\hat{Y}_M\}$ with their corresponding verbalized confidence $\{C_1, ...C_M\}$, and compute the confidence as follows:

$$C_{\text{conf}} = \frac{\sum_{i=1}^{M} \mathbb{I}\{\hat{Y}_i = \tilde{Y}\} \times C_i}{\sum_{i=1}^{M} C_i}. \tag{2}$$

**Pair-Rank.** This aggregation strategy is tailored for responses generated using the Top-K prompt, as it mainly utilizes the ranking information of the model's Top-K guesses. The underlying assumption is that the model's ranking between two options may be more accurate than the verbalized confidence it provides, especially given our observation that the latter tends to exhibit overconfidence.

Given a question with $N$ candidate responses, the $i$-th response consists of $K$ sequentially ordered answers, denoted as $\mathcal{S}_K^{(i)} = (S_1^{(i)}, S_2^{(i)}, \ldots, S_K^{(i)})$. Let $\mathcal{A}$ represent the set of unique answers across all $N$ responses, where $M$ is the total number of distinct answers. The event where the model ranks answer $S_u$ above $S_v$ (i.e., $S_u$ appears before $S_v$) in its $i$-th generation is represented as $(S_u \overset{(i)}{\succ} S_v)$. In contexts where the generation is implicit, this is simply denoted as $(S_u \succ S_v)$. Let $E_{uv}^{(i)}$ be the event where at least one of $S_u$ and $S_v$ appears in the $i$-th generation. Then the probability of $(S_u \succ S_v)$, conditional on $E_{uv}^{(i)}$ and a categorical distribution $P$, is expressed as $\mathbb{P}(S_u \succ S_v | P, E_{uv}^{(i)})$.

We then utilize a (conditional) maximum likelihood estimation (MLE) inspired approach to derive the categorical distribution $P$ that most accurately reflects these ranking events of all the $M$ responses:

$$\min_P -\sum_{i=1}^{N} \sum_{S_u \in \mathcal{A}} \sum_{S_v \in \mathcal{A}} \mathbb{I}\left\{S_u \overset{(i)}{\succ} S_v\right\} \cdot \log \mathbb{P}\left(S_u \succ S_v \mid P, E_{uv}^{(i)}\right) \quad \text{subject to} \sum_{S_u \in \mathcal{A}} P(S_u) = 1 \tag{3}$$

**Proposition 3.1.** *Suppose the Top-K answers are drawn from a categorical distribution $P$ without replacement. Define the event $(S_u \succ S_v)$ to indicate that the realization $S_u$ is observed before $S_v$ in the $i$-th draw without replacement. Under this setting, the conditional probability is given by:*

$$\mathbb{P}\left(S_u \succ S_v \mid P, E_{uv}^{(i)}\right) = \frac{P(S_u)}{P(S_u) + P(S_v)}$$

*The optimization objective to minimize the expected loss is then:*

$$\min_P -\sum_{i=1}^{N} \sum_{S_u \in \mathcal{A}} \sum_{S_v \in \mathcal{A}} \mathbb{I}\left\{S_u \overset{(i)}{\succ} S_v\right\} \cdot \log \frac{P(S_u)}{P(S_u) + P(S_v)} \quad s.t. \sum_{S_u \in \mathcal{A}} P(S_u) = 1 \tag{4}$$

To address this constrained optimization problem, we first introduce a change of variables by applying the softmax function to the unbounded domain. This transformation inherently satisfies the simplex constraints, converting our problem into an unconstrained optimization setting. Subsequently, optimization techniques such as gradient descent can be used to obtain the categorical distribution.

## 4 EXPERIMENT SETUP

**Datasets.** We evaluate the quality of confidence estimates across five types of reasoning tasks: 1) **Commonsense** Reasoning on two benchmarks, Sports Understanding (SportUND) (Kim, 2021) and StrategyQA (Geva et al., 2021) from BigBench (Ghazal et al., 2013); 2) **Arithmetic** Reasoning on two math problems, GSM8K (Cobbe et al., 2021) and SVAMP (Patel et al., 2021); 3) **Symbolic** Reasoning

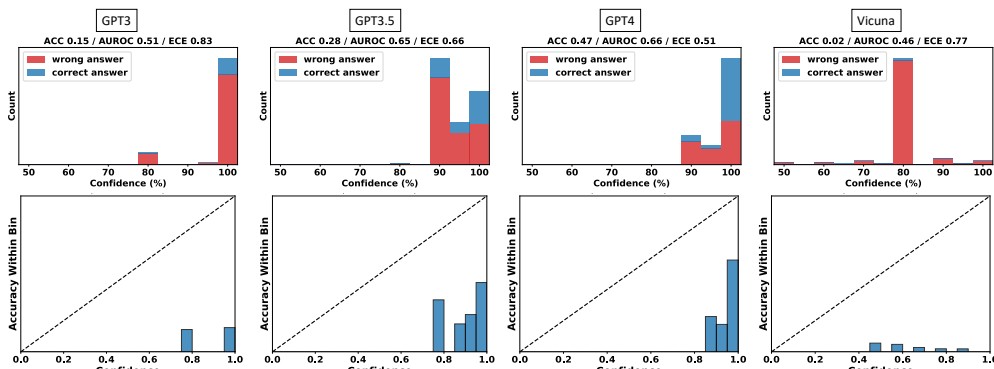

Figure 2: Empirical distribution (**First row**) and reliability diagram (**Second row**) of vanilla verbalized confidence across four models on GSM8K. The prompt used is in Table 14. From this figure, we can observe that 1) the confidence levels primarily range between 80% and 100%, often in multiples of 5; 2) the accuracy within each bin is much lower than its corresponding confidence, indicating significant overconfidence.

on two benchmarks, Date Understanding (DateUnd) (Wu & Wang, 2021) and Object Counting (ObjectCou) (Wang et al., 2019) from BigBench; 4) tasks requiring **Professional Knowledge**, such as Professional Law (Prf-Law) from MMLU (Hendrycks et al., 2021); 5) tasks that require **Ethical Knowledge**, e.g., Business Ethics (Biz-Ethics) from MMLU (Hendrycks et al., 2021).

**Models** We incorporate a range of widely used LLMs of different scales, including Vicuna 13B (Chiang et al., 2023), GPT-3 175B (Brown et al., 2020), GPT-3.5-turbo (OpenAI, 2021), GPT-4 (OpenAI, 2023) and LLaMA 2 70B (Touvron et al., 2023b).

**Evaluation Metrics.** To evaluate the quality of confidence outputs, two orthogonal tasks are typically employed: calibration and failure prediction (Naeini et al., 2015; Yuan et al., 2021; Xiong et al., 2022). Calibration evaluates how well a model's expressed confidence aligns with its actual accuracy: ideally, samples with an 80% confidence should have an accuracy of 80%. Such well-calibrated scores are crucial for applications including risk assessment. On the other hand, failure prediction gauges the model's capacity to assign higher confidence to correct predictions and lower to incorrect ones, aiming to determine if confidence scores can effectively distinguish between correct and incorrect predictions. In our study, we employ Expected Calibration Error (ECE) for calibration evaluation and Area Under the Receiver Operating Characteristic Curve (AUROC) for gauging failure prediction. Given the potential imbalance from varying accuracy levels, we also introduce AUPRC-Positive (PR-P) and AUPRC-Negative (PR-N) metrics to emphasize whether the model can identify incorrect and correct samples, respectively.

Further details on datasets, models, metrics, and implementation can be found in Appendix E.

## 5 EVALUATION AND ANALYSIS

To provide insights on the best practice for eliciting confidence, we systematically examine each component (see Figure 1) of the confidence elicitation framework (§3). We test the performance on eight datasets of five different reasoning types and five commonly used models (see §4), and yield the following key findings.

### 5.1 LLMs TEND TO BE OVERCONFIDENT WHEN VERBALIZING THEIR CONFIDENCE

**The distribution of verbalized confidences mimics how humans talk about confidence.** To examine model's capacity to express verbalized confidence, we first visualize the distribution of confidence in Figure 2. Detailed results on other datasets and models are provided in Appendix Figure 5. Notably, the models tend to have high confidence for all samples, appearing as multiples of 5 and with most values ranging between the 80% to 100% range, which is similar to the patterns identified in the training corpus for GPT-like models as discussed by Zhou et al. (2023). Such behavior suggests that models might be imitating human expressions when verbalizing confidence.

Table 2: **Vanilla Verbalized Confidence** of 4 models and 8 datasets (metrics are given by $\times 10^2$). Abbreviations are used: Date (Date Understanding), Count (Object Counting), Sport (Sport Understanding), Law (Professional Law), Ethics (Business Ethics). ECE > 0.25, AUROC, AUPRC-Positive, AUPRC-Negative < 0.6 denote significant deviation from ideal performance. Significant deviations in averages are highlighted in red. The prompt used is in Table 14.

| Metric | Model | GSM8K | SVAMP | Date | Count | Strategy | Sport | Law | Ethics | Avg |
|---|---|---|---|---|---|---|---|---|---|---|
| ECE ↓ | GPT-3 | 82.7 | 35.0 | 82.1 | 52.0 | 41.8 | 42.0 | 47.8 | 32.3 | 52.0 |
| | Vicuna | 76.0 | 70.7 | 17.0 | 45.3 | 42.5 | 37.5 | 45.2 | 34.6 | 46.1 |
| | LLaMA 2 | 71.8 | 36.4 | 38.5 | 58.0 | 26.2 | 38.8 | 42.2 | 36.5 | 43.6 |
| | GPT-3.5 | 66.0 | 22.4 | 47.0 | 47.1 | 26.0 | 25.1 | 44.3 | 23.4 | 37.7 |
| | GPT-4 | 31.0 | 10.7 | 18.0 | 26.8 | 16.1 | 15.4 | 17.3 | 8.5 | 18.0 |
| ROC ↑ | GPT3 | 51.2 | 51.7 | 50.2 | 50.0 | 49.3 | 55.3 | 46.5 | 56.1 | 51.3 |
| | Vicuna | 52.1 | 46.3 | 53.7 | 53.1 | 50.9 | 53.6 | 52.6 | 57.5 | 52.5 |
| | LLaMA 2 | 58.8 | 52.1 | 71.4 | 51.3 | 56.0 | 48.5 | 50.5 | 62.4 | 56.4 |
| | GPT-3.5 | 65.0 | 63.2 | 57.0 | 54.1 | 52.8 | 43.2 | 50.5 | 55.2 | 55.1 |
| | GPT4 | 81.0 | 56.7 | 68.0 | 52.0 | 55.3 | 60.0 | 60.9 | 68.0 | 62.7 |
| PR-N ↑ | GPT-3 | 85.0 | 37.3 | 82.2 | 52.0 | 42.0 | 46.4 | 51.2 | 41.2 | 54.7 |
| | Vicuna | 96.4 | 87.9 | 34.9 | 65.4 | 53.8 | 51.5 | 75.3 | 70.9 | 67.0 |
| | LLaMA 2 | 92.6 | 57.4 | 88.3 | 59.6 | 38.2 | 40.6 | 61.0 | 58.3 | 62.0 |
| | GPT-3.5 | 79.0 | 33.9 | 64.0 | 51.2 | 35.7 | 30.5 | 54.8 | 35.5 | 48.1 |
| | GPT-4 | 65.0 | 15.8 | 26.0 | 28.9 | 26.6 | 31.5 | 40.0 | 39.5 | 34.2 |
| PR-P ↑ | GPT-3 | 15.5 | 65.5 | 17.9 | 48.0 | 57.6 | 59.0 | 45.4 | 66.1 | 46.9 |
| | Vicuna | 4.10 | 11.0 | 69.1 | 39.1 | 47.5 | 52.0 | 27.2 | 38.8 | 36.1 |
| | LLaMA 2 | 11.9 | 46.3 | 46.6 | 41.4 | 68.6 | 58.3 | 39.2 | 65.0 | 47.2 |
| | GPT-3.5 | 38.0 | 81.3 | 57.0 | 54.4 | 67.2 | 67.5 | 45.8 | 70.5 | 60.2 |
| | GPT-4 | 57.0 | 90.1 | 88.0 | 73.8 | 78.6 | 79.3 | 73.4 | 87.2 | 78.4 |

**Calibration and failure prediction performance improve as model capacity scales.** The comparison of the performance of various models (Table 2) reveals a trend: as we move from GPT-3, Vicuna, GPT-3.5 to GPT-4, with the increase of model accuracy, there is also a noticeable decrease in ECE and increase in AUROC, e.g., approximate 22.2% improvement in AUROC from GPT-3 to GPT-4.

**Vanilla verbalized confidence exhibits significant overconfidence and poor failure prediction, casting doubts on its reliability.** Table 2 presents the performance of vanilla verbalized confidence across five models and eight tasks. According to the criteria given in Srivastava et al. (2023), GPT-3, GPT-3.5, and Vicuna exhibit notably high ECE values, e.g., the average ECE exceeding 0.377, suggesting that the verbalized confidence of these LLMs are poorly calibrated. While GPT-4 displays lower ECE, its AUROC and AUPRC-Negative scores remain suboptimal, with an average AUROC of merely 62.7%—close to the 50% random guess threshold—highlighting challenges in distinguishing correct from incorrect predictions.

## 5.2 HUMAN-INSPIRED PROMPTING STRATEGIES PARTIALLY REDUCE OVERCONFIDENCE

**Human-inspired prompting strategies improve model accuracy and calibration, albeit with diminishing returns in advanced models like GPT-4.** As illustrated in Figure 3, we compare the performance of five prompting strategies across five datasets on GPT-3.5 and GPT-4. Analyzing the average ECE, AUROC, and their respective performances within each dataset, human-inspired strategies offer consistent improvements in accuracy and calibration over the vanilla baseline, with modest advancements in failure prediction.

**No single prompting strategy consistently outperforms the others.** Figure 3 suggests that there is no single strategy that can consistently outperform the others across all the datasets and models. By evaluating the average rank and performance enhancement for each method over five task types, we find that *Self-Probing* maintains the most consistent advantage over the baseline on GPT-4, while *Top-K* emerges as the top performer on GPT-3.5.

**While ECE can be effectively reduced using suitable prompting strategies, failure prediction still remains a challenge.** Comparing the average calibration performance across datasets ('mean

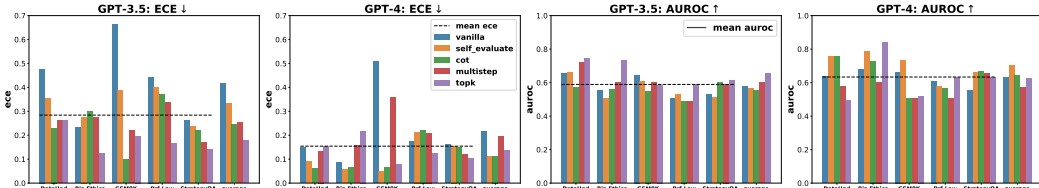

Figure 3: Comparative analysis of 5 prompting strategies over 5 datasets for 2 models (GPT-3.5 and GPT-4). The 'average' bar represents the mean ECE for a given prompting strategy across datasets. The 'mean ECE' line is the average across all strategies and datasets. AUROC is calculated in a similar manner. The accuracy comparison is shown in Appendix B.4.

ece' lines) and the average failure prediction performance ('mean auroc'), we find that while we can reduce ECE with the right prompting strategy, the model's failure prediction capability is still limited, i.e., close to the performance of random guess (AUROC=0.5). A closer look at individual dataset performances reveals that the proposed prompt strategies such as CoT have significantly increased the accuracy (see Table 8), while the confidence output distribution still remains at the range of $80\% - 100\%$, suggesting that *a reduction in overconfidence is due to the diminished gap between average confidence and accuracy, not necessarily indicating a substantial increase in the model's ability to judge the correctness of its responses*. For example, with the CoT prompting on the GSM8K dataset, GPT-4 with 93.6% accuracy achieves a near-optimal ECE 0.064 by assigning 100% confidence to all samples. However, since all samples receive the same confidence, it is challenging to distinguish between correct and incorrect samples based on the verbalized confidence.

## 5.3 Variance Among Multiple Responses Improves Failure Prediction

Table 3: Comparison of sampling strategies with the number of responses $M = 5$ on GPT-3.5. The prompt and aggregation strategies are fixed as CoT and Consistency when $M > 1$. To compare the effect of $M$, we also provide the baseline with $M = 1$ from Figure 3. Metrics are given by $\times 10^2$.

| Method | GSM8K | | Prf-Law | | DateUnd | | StrategyQA | | Biz-Ethics | | Average | |
|---|---|---|---|---|---|---|---|---|---|---|---|---|
| | ECE | AUROC | ECE | AUROC | ECE | AUROC | ECE | AUROC | ECE | AUROC | ECE | AUROC |
| Misleading (M=5) | 8.03 | 88.6 | 18.3 | 59.3 | 20.5 | 67.3 | 21.8 | **61.5** | 17.8 | 71.3 | **17.3** | 69.6 |
| Self-Random (M=5) | **6.28** | **92.7** | 26.0 | **65.6** | **17.0** | 66.8 | 23.3 | 60.8 | 20.7 | 79.0 | 18.7 | **73.0** |
| Prompt (M=5) | 35.2 | 74.4 | 31.5 | 60.8 | 23.9 | 69.8 | 16.1 | 61.3 | 15.0 | **79.5** | 24.3 | 69.2 |
| CoT (M=1) | 10.1 | 54.8 | 39.7 | 52.2 | 23.4 | 57.4 | 22.0 | 59.8 | 30.0 | 56.0 | 25.0 | 56.4 |
| Top-K (M=1) | 19.6 | 58.5 | **16.7** | 58.9 | 26.1 | **74.2** | **14.0** | 61.3 | **12.4** | 73.3 | 17.8 | 65.2 |

**Consistency among multiple responses is more effective in improving failure prediction and calibration compared to verbalized confidence ($M = 1$), with particularly notable improvements on the arithmetic task.** Table 3 demonstrates that the sampling strategy with 5 sampled responses paired with consistency aggregation consistently outperform verbalized confidence in calibration and failure prediction, particularly on arithmetic tasks, e.g., GSM8K showcases a remarkable improvement in AUROC from 54.8% (akin to random guessing) to 92.7%, effectively distinguishing between incorrect and correct answers. The average performance in the last two columns also indicates improved ECE and AUROC scores, suggesting that obtaining the variance among multiple responses can be a good indicator of uncertainty.

**As the number of sampled responses increases, model performance improves significantly and then converges.** Figure 7 exhibits the performance of various number of sampled responses $M$ from $M = 1$ to $M = 13$. The result suggests that the ECE and AUROC could be improved by sampling more responses, but the improvement becomes marginal as the number gets larger. Additionally, as the computational time and resources required for $M$ responses go linearly with the baseline ($M$=1), $M$ thus presents a trade-off between efficiency and effectiveness. Detailed experiments investigating the impact of the number of responses can be found in Appendix B.6 and B.7.

## 5.4 Introducing Verbalized Confidence Into The Aggregation Outperforms Consistency-only Aggregation

**Pair-Rank achieves better performance in calibration while Avg-Conf boosts more in failure prediction.** On the average scale, we find that Pair-Rank emerges as the superior choice for calibration that can reduce ECE to as low as 0.028, while Avg-Conf stands out for its efficacy in failure prediction.

Table 4: Performance comparison of aggregation strategies on GPT-4 using Top-K Prompt and Self-Random sampling. Pair-Rank aggregation achieves the lowest ECE in half of the datasets and maintains the lowest average ECE in calibration; Avg-Conf surpasses other methods in terms of AUROC in five out of the six datasets in failure prediction. Metrics are given by $\times 10^2$.

| Metric | Aggregator | GSM8K | Law | Date | Sport | Strategy | Ethics | Mean & Var |
|---|---|---|---|---|---|---|---|---|
| ECE ↓ | Consistency | **4.80** | 21.1 | **6.00** | 13.4 | 13.5 | 13.2 | 12.0 $_{\pm 0.3}$ |
| | Avg-Conf | 10.0 | **14.4** | 7.70 | 10.6 | 5.90 | 20.2 | 14.8 $_{\pm 0.7}$ |
| | Pair-Rank | 7.40 | 15.3 | 8.50 | **2.80** | **3.50** | **3.80** | **6.90** $_{\pm 0.2}$ |
| AUROC ↑ | Consistency | **84.4** | 66.2 | 68.9 | 60.3 | 65.4 | 56.3 | 66.9 $_{\pm 0.8}$ |
| | Avg-Conf | 41.0 | **68.0** | **72.7** | **64.8** | **70.5** | **84.4** | 66.9 $_{\pm 1.7}$ |
| | Pair-Rank | 80.3 | 66.5 | 67.4 | 61.9 | 62.1 | 67.6 | **67.6** $_{\pm 0.4}$ |

This observation agrees with the underlying principle that Pair-Rank learns the categorical distribution of potential answers through our $K$ observations, which aligns well with the notion of calibration and is therefore more likely to lead to a lower ECE. In contrast, Avg-Conf leverages the consistency, using verbalized confidence as a weighting factor for each answer. This approach is grounded in the observation that accurate samples often produce consistent outcomes, while incorrect ones yield various responses, leading to a low consistency. This assumption matches well with failure prediction, and is confirmed by the results in Table 4. In addition, our comparative analysis of various aggregation strategies reveals that introducing verbalized confidence into the aggregation (e.g., Pair-Rank and Avg-Conf) is more effective compared to consistency-only aggregation (e.g., Consistency), especially when LLM queries are costly, and we are limited in sampling frequency (set to $M = 5$ queries in our experiment). Verbalized confidence, albeit imprecise, reflects the model's uncertainty tendency and can enhance results when combined with ensemble methods.

## 6 DISCUSSIONS

In this study, we focus on confidence elicitation, i.e., empowering Large Language Models (LLMs) to accurately express the confidence in their responses. Recognizing the scarcity of existing literature on this topic, we define a systematic framework with three components: prompting, sampling and aggregation to explore confidence elicitation algorithms and then benchmark these algorithms on two tasks across eight datasets and five models. Our findings reveal that LLMs tend to exhibit overconfidence when verbalizing their confidence. This overconfidence can be mitigated to some extent by using proposed prompting strategies such as CoT and Self-Probing. Furthermore, sampling strategies paired with specific aggregators can improve failure prediction, especially in arithmetic datasets. We hope this work could serve as a foundation for future research in these directions.

**Comparative analysis of white-box and black-box methods.** While our method is centered on black-box settings, comparing it with white-box methods helps us understand the progress in the field. We conducted comparisons on five datasets with three white-box methods (see §B.1) and observed that although white-box methods indeed perform better, the gap is narrow, e.g., 0.522 to 0.605 in AUROC. This finding underscores that the field remains challenging and unresolved.

**Are current algorithms satisfactory?** Not quite. Our findings (Table 4) reveals that while the best-performing algorithms can reduce ECE to a quite low value like 0.028, they still face challenges in predicting incorrect predictions, especially in those tasks requiring professional knowledge, such as professional law. This underscores the need for ongoing research in confidence elicitation.

**What is the recommendation for practitioners?** Balancing between efficiency, simplicity, and effectiveness, and based on our empirical results, we recommend a stable-performing method for practitioners: **Top-K prompt + Self-Random sampling + Avg-Conf or Pair-Rank aggregation**. Please refer to Appendix D for the reasoning and detailed discussions, including the considerations when using black-box confidence elicitation algorithms and why these methods fail in certain cases.

**Limitations and Future Work:** *1) Scope of Datasets.* We mainly focuses on fixed-form and free-form question-answering QA tasks where the ground truth answer is unique, while leaving tasks such as summarization and open-ended QA to the future work. *2) Black-box Setting.* Our findings indicate black-box approaches remain suboptimal, while the white-box setting, with its richer information access, may be a more promising avenue. Integrating black-box methods with limited white-box access data, such as model logits provided by GPT-3, could be a promising direction.

## ACKNOWLEDGMENTS

This research is supported by the Ministry of Education, Singapore, under the Academic Research Fund Tier 1 (FY2023).

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

## A  PROOF OF PROPOSITION 3.1

**Notation.**  Given a question with $N$ candidate responses, the $i$-th response consists of $K$ sequentially ordered answers, denoted as $\mathcal{S}_K^{(i)} = (S_1^{(i)}, S_2^{(i)}, \ldots, S_K^{(i)})$. Let $\mathcal{A} = \{S_1, S_2, \ldots, S_M\}$ represent the set of unique answers across all $N$ responses, where $M$ is the total number of distinct answers. The event where the model ranks answer $S_u$ above $S_v$ in its $i$-th generation is represented as $(S_u \overset{(i)}{\succ} S_v)$. In contexts where the generation is implicit, this is simply denoted as $(S_u \succ S_v)$. Let $E_{uv}^{(i)}$ be the event where at least one of $S_u$ and $S_v$ appears in the $i$-th generation. The probability of $(S_u \succ S_v)$, given $E_{uv}^{(i)}$ and a categorical distribution $P$, is expressed as $\mathbb{P}(S_u \succ S_v | P, E_{uv}^{(i)})$.

**Proposition A.1.** *Suppose the Top-K answers are drawn from a categorical distribution $P$ without replacement. Define the event $(S_u \succ S_v)$ to indicate that the realization $S_u$ is observed before $S_v$ in the $i$-th draw without replacement. Under this setting, the conditional probability is given by:*

$$\mathbb{P}\left(S_u \succ S_v \mid P, E_{uv}^{(i)}\right) = \frac{P(S_u)}{P(S_u) + P(S_v)}$$

*The optimization objective to minimize the expected loss is then:*

$$\min_P -\sum_{i=1}^{N} \sum_{S_u \in \mathcal{A}} \sum_{S_v \in \mathcal{A}} \mathbb{I}\left\{S_u \overset{(i)}{\succ} S_v\right\} \cdot \log \frac{P(S_u)}{P(S_u) + P(S_v)} \quad s.t. \sum_{S_u \in \mathcal{A}} P(S_u) = 1 \qquad (5)$$

*Proof.* Let us begin by examining the position $j$ in the response sequence $\mathcal{S}_K^{(i)}$ where either $S_u$ or $S_v$ is first sampled, and the other has not yet been sampled. We denote this event as $F_j^{(i)}(S_u, S_v)$, and for simplicity, we refer to it as $F_j$:

$$\begin{aligned} F_j = F_j^{(i)}(S_u, S_v) &= \left\{\text{the earliest position in } \mathcal{S}_K^{(i)} \text{ where either } S_u \text{ or } S_v \text{ appears is } j\right\} \\ &= \left\{\forall m, n \in \{1, 2, ..., N\} \mid S_m^{(i)} = S_u, S_n^{(i)} = S_v, j = \min(m, n)\right\} \end{aligned} \qquad (6)$$

Given this event, the probability that $S_u$ is sampled before $S_v$ across all possible positions $j$ is:

$$\mathbb{P}(S_u \succ S_v \mid P, E_{uv}^{(i)}) = \sum_{j=1}^{N} \mathbb{P}(F_j \mid P, E_{uv}^{(i)}) \times \underbrace{\mathbb{P}(S_u \succ S_v \mid P, E_{uv}^{(i)}, F_j)}_{(a)} \qquad (7)$$

To further elucidate (1), which is conditioned on $F_j$, we note that the first sampled answer between $S_u$ and $S_v$ appears at position $j$. We then consider all potential answers sampled prior to $j$. For this, we introduce a permutation set $\mathcal{H}_{j-1}$ to encapsulate all feasible combinations of answers for the initial $j - 1$ samplings. A representative sampling sequence is given by: $\mathcal{S}_{j-1} = \{S_{(1)} \succ S_{(2)} \succ \cdots \succ S_{(j-1)} \mid \forall l \in \{1, 2, ..., j-1\}, S_{(l)} \in \mathcal{A} \setminus \{S_u, S_v\}\}$.

Consequently, (a) can be articulated as:

$$\mathbb{P}(S_u \succ S_v \mid P, E_{uv}^{(i)}, F_j) = \sum_{\mathcal{S}_{j-1} \in \mathcal{H}_{j-1}} \mathbb{P}(\mathcal{S}_{j-1} \mid P, E_{uv}^{(i)}, F_j) \times \underbrace{\mathbb{P}(S_u \succ S_v \mid P, E_{uv}^{(i)}, \mathcal{S}_{j-1}, F_j)}_{(b)} \qquad (8)$$

Consider the term (b), which signifies the probability that, given the first $j - 1$ samplings and the restriction that the $j$-th sampling can only be $S_u$ or $S_v$, $S_u$ is sampled prior to $S_v$. This probability is articulated as:

$$\begin{aligned} \mathbb{P}(S_u \succ S_v \mid P, E_{uv}^{(i)}, F_j, \mathcal{S}_{j-1}) &= \frac{\mathbb{P}(S_j^{(i)} = S_u \mid P, E_{uv}^{(i)}, F_j, \mathcal{S}_{j-1})}{\mathbb{P}(S_j^{(i)} = S_u \mid P, E_{uv}^{(i)}, F_j, \mathcal{S}_{j-1}) + \mathbb{P}(S_j^{(i)} = S_v \mid P, E_{uv}^{(i)}, F_j, \mathcal{S}_{j-1})} \\ &= \frac{\frac{P(S_u)}{1 - \sum_{S_m \in \mathcal{S}_{j-1}} P(S_m)}}{\frac{P(S_v)}{1 - \sum_{S_m \in \mathcal{S}_{j-1}} P(S_m)} + \frac{P(S_u)}{1 - \sum_{S_m \in \mathcal{S}_{j-1}} P(S_m)}} \\ &= \frac{P(S_u)}{P(S_u) + P(S_v)} \end{aligned}$$

$$(9)$$

Integrating equation (9) into equation (8), we obtain:

$$
\begin{aligned}
\mathbb{P}(S_u \succ S_v \mid P, E_{uv}^{(i)}, F_j) &= \sum_{\mathcal{S}_{j-1} \in \mathcal{H}_{j-1}} \mathbb{P}(\mathcal{S}_{j-1} \mid P, F_j) \times \frac{P(S_u)}{P(S_u) + P(S_v)} \\
&= \frac{P(S_u)}{P(S_u) + P(S_v)} \times \sum_{\mathcal{S}_{j-1} \in \mathcal{H}_{j-1}} \mathbb{P}(\mathcal{S}_{j-1} \mid P, E_{uv}^{(i)}, F_j) \qquad (10) \\
&\overset{(c)}{=} \frac{P(S_u)}{P(S_u) + P(S_v)}
\end{aligned}
$$

Subsequently, incorporating equation (10) into equation (7), we deduce:

$$
\begin{aligned}
\mathbb{P}(S_u \succ S_v \mid P, E_{uv}^{(i)}) &= \sum_{j=1}^{K} \mathbb{P}(F_j \mid P, E_{uv}^{(i)}) \times \frac{P(S_u)}{P(S_u) + P(S_v)} \\
&= \frac{P(S_u)}{P(S_u) + P(S_v)} \times \sum_{j=1}^{K} \mathbb{P}(F_j \mid P, E_{uv}^{(i)}) \qquad (11) \\
&\overset{(d)}{=} \frac{P(S_u)}{P(S_u) + P(S_v)}
\end{aligned}
$$

The derivations in (c) and (d) employ the Law of Total Probability.

Incorporating Equation 11 into Equation 3, the minimization objective is formulated as:

$$
\min_{P} - \sum_{i=1}^{N} \sum_{S_u \in \mathcal{A}} \sum_{S_v \in \mathcal{A}} \mathbb{I}\{S_u \overset{(i)}{\succ} S_v\} \times \log \frac{P(S_u)}{P(S_u) + P(S_v)} \quad \text{s.t.} \sum_{S_u \in \mathcal{A}} P(S_u) = 1 \qquad (12)
$$

$\square$

## B    DETAILED EXPERIMENT RESULTS

### B.1    WHITE-BOX METHODS OUTPERFORM BLACK-BOX METHODS, BUT THE GAP IS NARROW.

**Comparative Analysis of White-Box and Black-Box Methods**: Which performs better - white-box or black-box methods? Do white-box methods, with their access to more internal information, outperform their black-box counterparts? If so, how large is the performance gap? To address these questions, we conduct a comparative analysis of white-box methods based on token probability against black-box models utilizing verbalized confidence.

**Implementation details**: We utilize the probabilities of each output token to develop three token-probability-based white-box methods: 1) **Sequence Probability (seq-prob)**, which aggregates the probabilities of all tokens; 2) **Length-Normalized Sequence Probability (len-norm-prob)**, which normalizes the sequence probability based on sequence length, i.e., seq-prob$^{1/\text{length}}$; 3) **Key Token Probability (token-prob)**, designed to focus on the result-specific tokens, e.g., "35" in the output sequence "Explanation: ....; Answer: 35; ...", thereby minimizing the influence of irrelevant output tokens. For our implementation, we use the Chain-of-Thought and Top-K Verbalized Confidence prompt to acquire verbalized confidence and select GPT3 as the backbone model.

**Findings:** Our comparative analysis, detailed in Table 5 and Table 6, yields several key insights: 1) Generally, **white-box methods exhibit better performance**, with length-normalized sequence probability and key token probability emerging as the most effective methods across five datasets and four evaluation metrics. 2) **The gap between white-box and black-box methods is relatively modest**. Moreover, even the best-performing **white-box methods fall short of achieving satisfactory results**. This is particularly apparent in the AUROC metric, where the performance of nearly all methods across various datasets ranges between 0.5-0.6, signifying a limited capability in distinguishing between correct and incorrect responses. 3) These experimental results suggest that **uncertainty estimation in LLMs remains a challenging and unresolved issue**. As mentioned in our introduction, the logit-based methods, which predominantly capture the model's uncertainty regarding the next token, are less effective in capturing the semantic uncertainty inherent in their textual meanings. Although several alternative approaches like semantic uncertainty (Kuhn et al., 2023) have been proposed, they come with significant computational demands. This scenario underscores the need for future research on both white-box and black-box methods to discover more efficient and effective methods for uncertainty estimation in LLMs.

### B.2    HOW MUCH DOES THE ROLE-PLAY PROMPT AFFECT THE PERFORMANCE?

To explore how the verbalized confidence elicitation performance varies when LLMs are asked to play different personalities such as *"confident"* and *"cautious"*, we conduct the experiment in Figure 4 and in Table 7. The results are derived when adding "You are a confident GPT" (**Left**) and "You are a cautious GPT" (**Right**) to the beginning of the Chain of Thought (CoT) prompt (Table 15). The experimental results show that the difference between their confidence distribution seems minimal, suggesting that assuming different personalities does not significantly affect performance metrics such as accuracy, ECE, and AUROC.

### B.3    HOW IS THE DISTRIBUTION OF VANILLA VERBALIZED CONFIDENCE ACROSS MODELS AND DATASETS?

Figure 5 presents the empirical distribution of vanilla verbalized confidence across 4 models and 5 datasets. Notably, all the models output confidence as the multiples of 5, with most values ranging between the 80% to 100% range. This behavior resembles the patterns identified in the training corpus for GPT-like models as discussed by Zhou et al. (2023). Such behavior suggests that models might be imitating human expressions when verbalizing confidence.

### B.4    DETAILED PERFORMANCE OF DIFFERENT PROMPTING STRATEGIES

**Multi-step and Top-K prompting strategies demonstrate promising results in reducing ECE and improving AUROC, with Top-K being relatively more effective.** Figure 6 presents a comparison of various prompting strategies (CoT, Multi-Step, Top-K) against vanilla verbalized confidence.

Table 5: Performance comparison (metrics are given by $\times 10^2$) of token-probability-based white-box methods including the baseline sequence probability ("seq-prob"), length-normalized sequence probability ("len-norm-prob") and key token probability ("token-prob"), and black-box verbalized confidence ("Verbalized") on GPT-3 using Top-K Prompt.

| Dataset | Acc | Method | ECE | AUROC | AUPRC-P | AUPRC-N |
|---|---|---|---|---|---|---|
| StrategyQA | 59.90 | Verbalized | 39.04 | 50.34 | 60.06 | 40.27 |
| | | seq-prob | **7.14** | 55.50 | 62.99 | 45.22 |
| | | len-norm-prob | 37.65 | 55.50 | 62.99 | 45.22 |
| | | token-prob | 32.43 | **60.61** | **69.90** | **47.10** |
| Biz-Ethics | 61.00 | Verbalized | **18.20** | 66.27 | 71.95 | 50.59 |
| | | seq-prob | 48.49 | 62.30 | 71.07 | 52.23 |
| | | len-norm-prob | 33.70 | 62.30 | 71.07 | 52.23 |
| | | token-prob | 27.65 | **67.00** | **74.89** | **55.01** |
| GSM8K | 11.52 | Verbalized | 77.40 | 54.05 | 12.70 | 89.01 |
| | | seq-prob | **7.73** | 69.80 | 20.40 | 94.71 |
| | | len-norm-prob | 72.41 | **70.61** | **21.23** | **94.75** |
| | | token-prob | 35.60 | 69.29 | 20.63 | 94.27 |
| DateUND | 15.72 | Verbalized | 83.47 | 50.80 | 15.93 | 84.54 |
| | | seq-prob | **16.10** | **62.93** | **22.39** | **90.61** |
| | | len-norm-prob | 81.27 | **62.93** | **22.39** | **90.61** |
| | | token-prob | 74.19 | 54.25 | 19.28 | 83.85 |
| Prf-Law | 44.92 | Verbalized | 41.55 | 49.54 | 44.43 | 55.78 |
| | | seq-prob | **32.31** | 51.07 | 45.75 | 56.70 |
| | | len-norm-prob | 49.66 | 51.06 | 45.75 | 56.79 |
| | | token-prob | 43.26 | **61.24** | **53.84** | **64.69** |

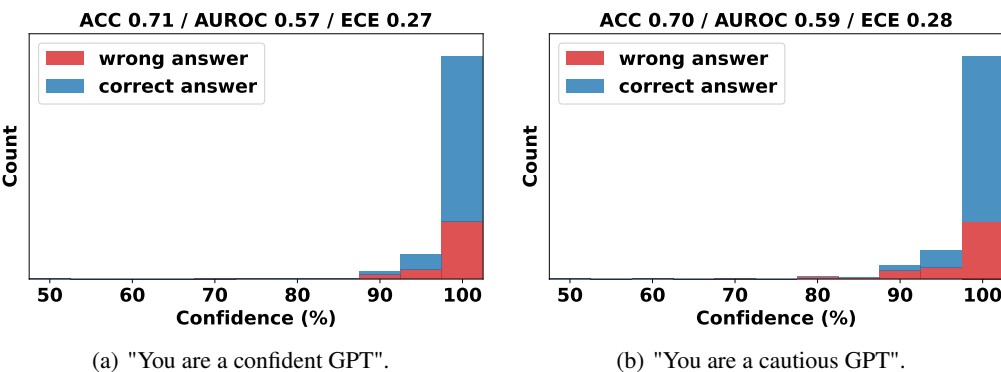

(a) "You are a confident GPT".    (b) "You are a cautious GPT".

Figure 4: Distribution of the verbalized confidence with different specified role descriptions in prompts. The results are derived when adding "You are a confident GPT" (**Left**) and "You are a cautious GPT" (**Right**) to the beginning of the Chain of Thought (CoT) prompt (Table 15). All other aspects of the prompts remain identical to the standard CoT format.

The detailed performance of CoT, Multi-Step, and Top-K prompt can be found in Table 8, Table 9 and Table 10, respectively. Judging from the 'average' bar, which computes the mean value across five datasets, both Multi-step and Top-K prompting strategies effectively reduce ECE and enhance AUROC. Moreover, Top-K shows relatively better performance improvements. The intuition behind this improvement is that this prompting strategy, requesting the model to generate multiple guesses along with their corresponding confidences, naturally nudges the model to be aware of the existence of

Table 6: Performance comparison (metrics are given by $\times 10^2$) of token-probability-based white-box methods including the baseline sequence probability ("seq-prob"), length-normalized sequence probability ("len-norm-prob") and key token probability ("token-prob"), and black-box verbalized confidence ("Verbalized") on GPT-3 using CoT Prompt.

| Dataset | Acc | Method | ECE | AUROC | AUPRC-P | AUPRC-N |
|---------|-----|--------|-----|-------|---------|---------|
| DateUND | 62.33 | Verbalized | 37.40 | 50.36 | 62.50 | 38.12 |
| | | seq-prob | 62.30 | 56.37 | 65.14 | 43.21 |
| | | len-norm-prob | **15.78** | **58.70** | **66.57** | **47.24** |
| | | token-prob | 27.32 | 40.27 | 55.20 | 35.69 |
| StrategyQA | 67.57 | Verbalized | 29.74 | 51.37 | 68.16 | 34.54 |
| | | seq-prob | 67.56 | 52.04 | 69.58 | 33.48 |
| | | len-norm-prob | **6.79** | 52.11 | **70.41** | 33.43 |
| | | token-prob | 30.59 | **53.00** | 68.80 | **36.89** |
| Biz-Ethics | 59.00 | Verbalized | 40.90 | 49.15 | 58.59 | 41.00 |
| | | seq-prob | **26.50** | 58.99 | 64.30 | 47.45 |
| | | len-norm-prob | 39.43 | 58.99 | 64.30 | 47.45 |
| | | token-prob | 36.31 | **67.38** | **75.33** | **54.89** |
| GSM8K | 52.31 | Verbalized | 47.49 | 50.32 | 52.47 | 48.02 |
| | | seq-prob | 52.30 | 57.47 | 56.75 | 54.39 |
| | | len-norm-prob | **29.80** | 57.92 | **58.84** | 55.23 |
| | | token-prob | 44.94 | **58.44** | 57.54 | **60.43** |
| Prf-Law | 44.85 | Verbalized | 53.43 | 50.13 | 44.90 | 55.91 |
| | | seq-prob | 44.85 | 51.88 | 46.62 | 56.09 |
| | | len-norm-prob | **31.00** | 50.10 | 45.34 | 55.32 |
| | | token-prob | 51.75 | **57.83** | **50.53** | **62.52** |

| Role | Model | ACC | ECE | AUROC | AUPRC-P | AUPRC-N |
|------|-------|-----|-----|-------|---------|---------|
| Confident | chatgpt-0613 | 0.7103 | 0.2741 | 0.5679 | 0.7398 | 0.3635 |
| Cautious | chatgpt-0613 | 0.6983 | 0.2812 | 0.5946 | 0.7415 | 0.4009 |

Table 7: Performance Comparison of Verbalized Confidence Elicitation with two types of prompt: "You are a confident GPT" and "You are a cautious GPT". The difference between these two prompts seems minimal, suggesting that asking LLMs to take on different personae does not significantly affect the performance.

various possible answers, preventing overconfidence in a single response and promoting re-evaluation of given answers.

## B.5   TOP-K VERBALIZED CONFIDENCE PERFORMANCE

The detailed experiments performance of Top-K verbalized confidence can be found in Table 10.

## B.6   IMPACT OF MISLEADING PROMPTS IN MISLEADING SAMPLING STRATEGY

To study the effect of different misleading prompts used in misleading sampling strategy, we categorize the prompts into three types: Weak Claim, Strong Claim, and External Source. The specific prompts are shown in Table 11. The Weak Claim category represents prompts that exhibit uncertainty, such as "I vaguely remember the answer is" or "I think the answer should be". These prompts suggest that the user is providing misleading information to the Large Language Model (LLM) but lacks confidence in the information provided. The Strong Claim category includes prompts like "I am pretty sure that this is" or "I am very confident that", which express a high degree of certainty. The External Source

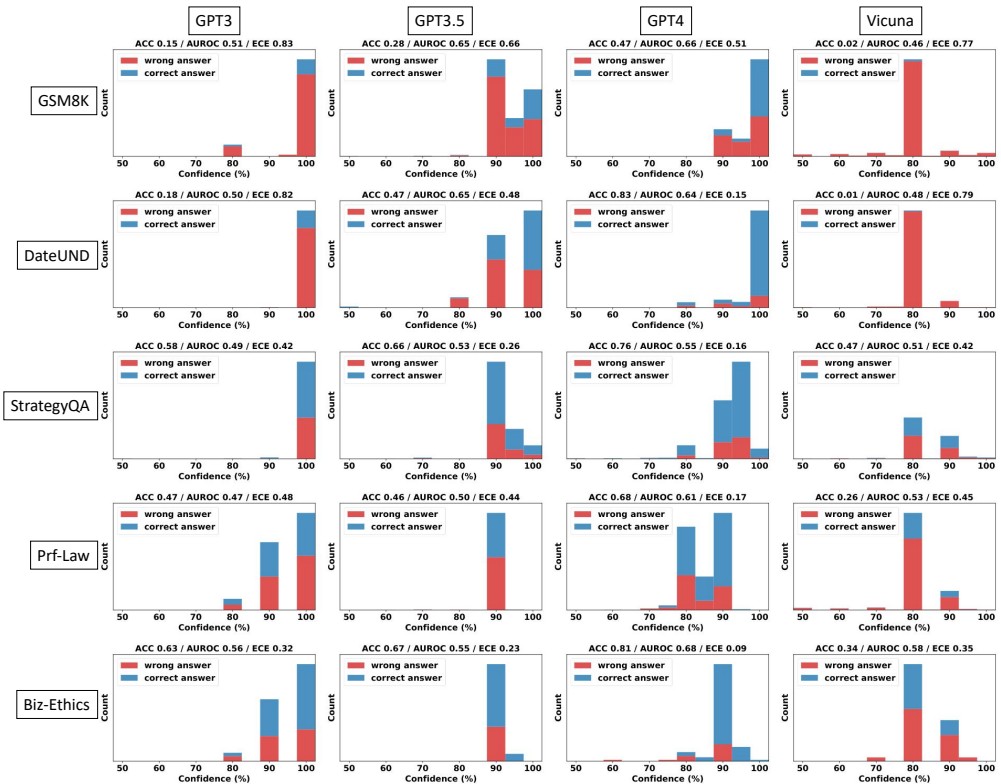

Figure 5: Empirical distribution of vanilla verbalized confidence across 4 models and 5 datasets. The prompt used is in Table 14. From this figure, we can observe that 1) the confidence levels primarily range between 80% and 100%, often in multiples of 5; 2) a large portion of incorrect predictions (red) has been observed even in the 100% confidence bar, indicating significant overconfidence.

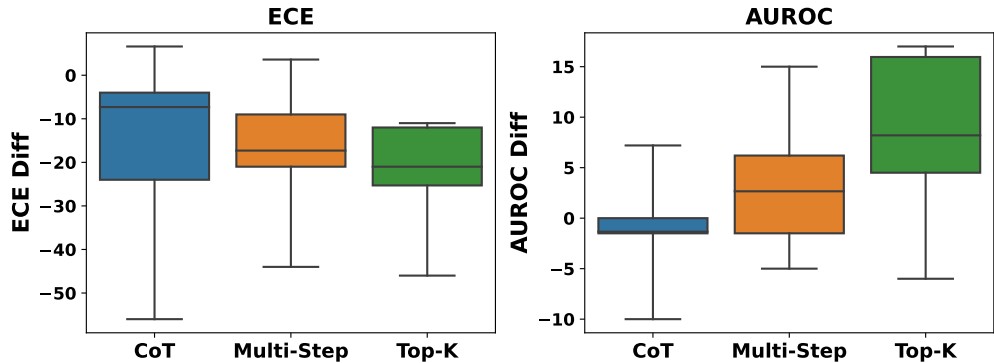

Figure 6: Performance Comparison of four verbalized confidence methods: vanilla, CoT, Multi-Step, Top-K in terms of ECE and AUROC for five types of datasets on GPT-3.5. Refer to Table 10 for detailed results.

category represents prompts that cite external sources as their evidence, such as "Wikipedia says" or "the latest research shows that".

Our experimental results (Table 11) indicate that the Weak Claim category performs better. A possible explanation is that on one hand even providing weak misleading information, the model will analyze and reassess their answers. On the other hand, since the misleading answers are generated randomly, confidently providing this information can sometimes lead to negative effects. For example, the model provides a correct answer with moderate confidence. However, if a misleading hint is provided

Table 8: Improvement of verbalized confidence with Chain-of-Thought Prompts

| Dataset | CoT | GPT3.5 | | |
|---------|-----|--------|-----|-------|
| | | ACC(%) | ECE | AUROC |
| GSM8K | ✗ | 28 | 66 | 65 |
| | ✓ | 80.3 | 10 | 55 |
| DateUnd | ✗ | 47 | 48 | 65 |
| | ✓ | 73.2 | 23 | 57 |
| StrategyQA | ✗ | 65.8 | 26 | 53 |
| | ✓ | 67.9 | 22 | 60 |
| Prf-Law | ✗ | 45.5 | 44 | 50 |
| | ✓ | 51.7 | 37 | 49 |
| Biz-Ethics | ✗ | 67 | 23 | 55 |
| | ✓ | 61 | 30 | 56 |

Table 9: Evaluation of multistep verbalized confidence for GPT-3.5 Models

| Dataset | SA | GPT3.5 | | |
|---------|-----|--------|-----|-------|
| | | ACC(%) | ECE | AUROC |
| GSM8K | ✗ | 80.3 | 10 | 55 |
| | ✓ | 76.2 | 22 | 60 |
| DateUnd | ✗ | 73.2 | 23 | 57 |
| | ✓ | 63.6 | 26 | 72 |
| StrategyQA | ✗ | 67.9 | 22 | 60 |
| | ✓ | 68.7 | 17 | 59 |
| Prf-Law | ✗ | 51.7 | 37 | 49 |
| | ✓ | 49.6 | 27 | 49 |
| Biz-Ethics | ✗ | 61 | 30 | 56 |
| | ✓ | 61.6 | 27 | 60 |

with high confidence or is supported by an external source, the model may be inclined to believe the prompt and alter its predictions.

### B.7 IMPACT OF THE NUMBER OF CANDIDATE ANSWERS

We investigate the impact of the number of candidate answers, denoted as $K$, utilized in the sampling strategy. Specifically, $K$ represents the number of queries used to construct the set of candidate answers for consistency calculation. We illustrate its calibration performance (ECE) and failure prediction performance (AUROC) in relation to varying numbers of $K$ (ranging from $K = 1$ to $K = 13$) in Figure 7.

The results indicate that, in terms of AUROC, a higher candidate set size $K$ contributes to superior performance and reduced variance. However, the optimal candidate size $K$ for ECE varies across different datasets. For instance, the StrategyQA dataset exhibits improved performance with a larger $K$, whereas the Business Ethics dataset generally performs better with a moderate number of candidate answers (e.g., $K = 4$). This observation can be attributed to the limited variability of misleading information (restricted to 4 types) used in our experiments for the Business Ethics dataset, implying that the introduction of a large number of more queries does not significantly enhance the information pool. Therefore, to strike a balance between computational efficiency and performance, we set the candidate set to be 4 in our study.

Table 10: Evaluation of Top-K verbalized confidence on GPT-3.5.

| Dataset | GPT3.5 | | |
|---|---|---|---|
| | ACC(%) | ECE | AUROC |
| GSM8K | 22.8 | 19.6 | 58.5 |
| DateUnd | 33.3 | 26.1 | 74.2 |
| StrategyQA | 61.3 | 14 | 61.3 |
| Prf-Law | 42.2 | 16.7 | 58.9 |
| Biz-Ethics | 67.0 | 12.4 | 73.3 |

Table 11: Different Prompts used for misleading sampling strategy.

| Group | Consistency Confidence Misleading Prompts |
|---|---|
| Weak Claim | Hint: I think the answer should be |
| Weak Claim | Hint: I read online the answer is |
| Weak Claim | Hint: I vaguely remember the answer is |
| Externel Source | Hint: My teacher tells me the answer is |
| Externel Source | Hint: The textbook shows that answer is |
| Externel Source | Hint: An expert says that the answer is |
| Externel Source | Hint: Latest research shows that the answer is |
| Externel Source | Hint: Wikipedia claims that the answer is |
| Strong Claim | Hint: I am 80% sure the answer is |
| Strong Claim | Hint: I am pretty sure the answer is |
| Strong Claim | Hint: Undoubtedly the answer is |
| Strong Claim | Hint: With my 100% confidence the answer is |

## B.8 Performance of different confidence elicitation methods

## C Related Works

**Confidence Elicitation in LLMs.** Confidence elicitation refers to the process of estimating LLM's confidence in their responses, without relying on model fine-tuning or accessing the proprietary information of LLMs. Within this scope, Lin et al. (2022) proposes the concept of verbalized confidence that elicits the model to output confidence directly. However, the evaluation is tailored

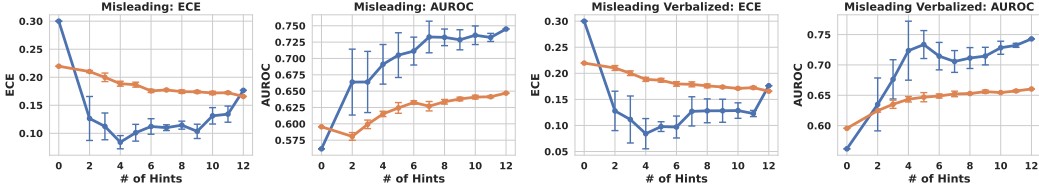

Figure 7: Impact of the number of responses responses on GPT-3.5. The sampling strategy is fixed as misleading. For every given number of misleading hints, we randomly sample the specified number of queries for 5 times and calculate the mean ECE and AUROC, and compute its variance(plotted as error bar). Note that the number of hints plus 1 is the number of responses sampled during experiment.

Table 12: The performance of varying prompt groups in StrategyQA on GPT-3.5. The group exhibiting the optimal performance is emphasized in bold. The experimental results indicate that the Weak Claim category performs better.

| Method | Hint Group | GPT-3.5 | |
|---|---|---|---|
| | | ECE | AUROC |
| Induced Consistency Confidence | Weak Claim | **19.7** | **62.0** |
| | Strong Claim | 19.5 | 61.4 |
| | External Source | 18.2 | 60.8 |
| verbalized-consistency confidence | Weak Claim | **19.8** | **65.4** |
| | Strong Claim | 19.5 | 64.6 |
| | External Source | 18.2 | 63.4 |

for pretrained language models that are fine-tuned on specific datasets, and its zero-shot verbalized confidence remains unexplored. Mielke et al. (2022) proposes to train an external calibrator while relies on model representations that are not readily accessible. Zhou et al. (2023) examine the impact of confidence in prompts but does not directly provide confidence to users. Our work aligns most closely with the concurrent study by Tian et al. (2023), which also focuses on the use of prompting strategies. However, our approach diverges by aiming to explore a broader method space, introducing a unified framework consisting of three components and conducting a systematic evaluation of strategies within each. The Top-K method, as proposed in (Tian et al., 2023), serves as an instance within our framework, and its performance can be augmented when integrated with other strategies from our framework. Furthermore, our investigation extends beyond the RLHF-LMs primarily analyzed in the concurrent study, and encompasses a broader spectrum of models. This allows us to probe the implications of different model sizes and structures. Our findings also underscore that all existing methods still face challenges with more complex tasks, contributing to a more holistic understanding of confidence elicitation in the field.

**Calibration.** Modern neural networks are shown to be poorly calibrated, often manifesting overconfidence (Guo et al., 2017; Minderer et al., 2021; Xiong et al., 2023). Calibration seeks to address the issue by aligning the model's confidence with the accuracy of samples within the same confidence level (Guo et al., 2017; Minderer et al., 2021). To achieve this, a variety of methods have been proposed, which can be broadly divided into scaling-based methods (Guo et al., 2017; Deng et al., 2023; Zhang et al., 2020) and binning-based methods (Zadrozny & Elkan, 2001; Zhang et al., 2020). Within the scope of LLMs, Jiang et al. (2021) investigates the calibration of generative language models (T5, BART, and GPT-2) and discovers that these models' probabilities on question-answering tasks are poorly calibrated. Similarly, Chen et al. (2022) finds that PLMs are not well calibrated and pretraining improves model calibration. On the other hand, Kadavath et al. (2022) studies the calibration of LLMs (parameter size ranging 800M to 50B), finding that larger models appear to be well-calibrated on multiple choice and true/false questions when provided in the right format. However, these evaluations mainly focus on the probabilities derived from logits, which are unavailable for closed-source LLMs like GPT-4. This also motivates us to study confidence elicitation methods that do not require model fine-tuning or access to model logits or embeddings.

# D    BEST PRACTICE AND RECOMMENDATIONS FOR PRACTITIONERS

## D.1    WHAT IS THE RECOMMENDATION FOR PRACTITIONERS?

Balancing between efficiency, simplicity, and effectiveness, we recommend a stable-performing method from our empirical results as advice for practitioners: **Top-K prompt + Self-Random sampling + Avg-Conf or Pair-Rank aggregation**. The recommendation is based on: 1) Top-K outperforms all other methods on GPT-3.5 and is comparable to the top-performing method Self-Probing on GPT4. Compared to Self-Probing which requires two inference phases, the Top-K prompt is chosen for the balance between effectiveness and efficiency. 1) As shown in Sec 5.3, ensemble

Table 13: Performance of different confidence elicitation methods: verbalize-based (Top-K and CoT Verbalized Confidence), consistency-based (Self-Consistency and Induced consistency), and their hybrid combinations. The best-performing method for each dataset is highlighted in **bold**.

| Metric | Method | GSM8K | DateUND | StrategyQA | Prf-Law | Biz-Ethics | Avg |
|---|---|---|---|---|---|---|---|
| ECE ↓ | Top-K (M=1) | 39.8 | 40.1 | **14.0** | 16.7 | **12.4** | 24.6 |
| | CoT (M=1) | 10.1 | 23.4 | 22.0 | 39.7 | 30.0 | 25.0 |
| | Self-Random+Consistency (M=5) | **6.28** | 17.0 | 23.3 | 26.0 | 20.7 | 18.7 |
| | Misleading + Cons (M=5) | 8.03 | 20.5 | 21.8 | 18.3 | 17.8 | 17.3 |
| | Self-Random + Avg-Conf (M=5) | 9.28 | **14.6** | 15.9 | 18.3 | 15.8 | 14.8 |
| | Misleading + Avg-Conf (M=5) | 7.40 | 17.6 | 15.0 | **12.8** | 18.2 | **14.2** |
| ROC ↑ | Top-K (M=1) | 59.9 | **76.3** | 61.3 | 58.9 | 73.3 | 65.9 |
| | CoT (M=1) | 54.8 | 57.4 | 59.8 | 52.2 | 56.0 | 56.4 |
| | Self-Random+Consistency (M=5) | **92.7** | 66.8 | 60.8 | **65.6** | 79.0 | 73.0 |
| | Misleading + Cons (M=5) | 88.6 | 67.3 | 61.5 | 59.3 | 71.3 | 69.6 |
| | Self-Random + Avg-Conf (M=5) | 92.5 | 68.8 | **66.2** | 65.3 | **79.5** | **74.5** |
| | Misleading + Avg-Conf (M=5) | 88.8 | 63.8 | 65.6 | 60.4 | 72.4 | 70.2 |
| PR-P ↑ | Top-K (M=1) | 27.7 | 62.8 | 68.4 | 49.3 | 82.2 | 58.1 |
| | CoT (M=1) | 81.8 | 76.6 | 72.8 | 49.2 | 64.3 | 68.9 |
| | Self-Random+Consistency (M=5) | 96.9 | 81.0 | 73.7 | 59.4 | 82.3 | 78.7 |
| | Misleading + Cons (M=5) | 95.1 | 81.0 | 74.1 | 54.7 | 77.6 | 76.5 |
| | Self-Random + Avg-Conf (M=5) | **97.0** | **84.4** | 78.3 | **60.3** | **83.1** | **80.6** |
| | Misleading + Avg-Conf (M=5) | 95.3 | 79.0 | **79.1** | 56.4 | 80.9 | 78.1 |
| PR-N ↑ | Top-K (M=1) | 80.2 | **79.8** | 45.7 | 56.0 | 50.7 | **62.5** |
| | CoT (M=1) | 23.1 | 30.7 | 40.5 | 53.9 | 43.7 | 38.4 |
| | Self-Random+Consistency (M=5) | 79.7 | 44.6 | 39.5 | 63.8 | 63.4 | 58.2 |
| | Misleading + Cons (M=5) | 71.2 | 44.2 | 41.3 | 58.7 | 55.1 | 54.1 |
| | Self-Random + Avg-Conf (M=5) | **81.5** | 51.8 | **45.8** | **65.3** | **64.9** | 61.9 |
| | Misleading + Avg-Conf (M=5) | 73.5 | 42.4 | 45.4 | 60.9 | 57.1 | 55.9 |

methods (e.g., $M = 5$) are consistently more effective than verbalized confidence ($M = 1$) in eliciting a model's confidence. Regarding the sampling strategies, Self-Random is selected for being more straightforward and commonly used, since the performance difference of different sampling strategies is minimal.. 3) For aggregation, strategies based on both answers and verbalized confidences (e.g., Avg-Conf and Pair-Rank) outperform *aggregation based on answers only (e.g., consistency)*. Then we recommend Pair-Rank and Avg-Conf for different downstream tasks according to their relatively good performance on different metrics. For example, for tasks that prioritize the exact confidence values, like calculating expected risk, Pair-Rank is recommended, while Avg-Conf is better suited for tasks related to failure prediction, e.g., factual error detection. Additionally, it is noteworthy that using Top-K alone does not improve accuracy as much as Chain of Thought (CoT), but the use of ensemble methods compensates for this.

### D.2 WHAT ARE THE CONSIDERATIONS WHEN USING BLACK-BOX CONFIDENCE ELICITATION ALGORITHMS?

Careful consideration is necessary due to significant limitations: 1) The reliability of the given confidence must be assessed by considering multiple metrics, such as both ECE and AUROC. As discussed in section 5.2, a high ECE does not imply that the model's outputs accurately represent model correctness. Metrics including AUROC and detailed information such as the confidence distribution plot should also be considered for a comprehensive evaluation and better understanding. 2) LLMs are not explicitly modeled to express uncertainty in textual outputs, and descriptions of uncertainty in the training corpus are mostly human expressions, which are often considered inaccurate (Garthwaite et al., 2005b). Dependence on such confidence for real-world applications requires careful checking, especially given the consistently high confidence levels shown in Figure 2, no matter whether the question is correctly answered or not.

### D.3 DISCUSSIONS ON WHY SOME STRATEGIES WORK, AND WHY SOME DO NOT WORK

In this section, we discuss the effective strategies and analyze the rationale behind these mechanisms.

**Sampling**    Consistency among multiple responses is more effective compared to verbalized confidence ($M = 1$), with particularly notable improvements on the arithmetic task. This is because sampling more queries allows us to directly approximate the model's internal distribution, $P_{model}(\mathbf{x}_t|\mathbf{x}_{1:t-1})$, which is trained to mirror the ground truth data distribution. Issues making this method ineffective can be: 1) the model's poor calibration (Kuhn et al., 2023), i.e., $P_{model}(\mathbf{x}_t|\mathbf{x}_{1:t-1})$ does not align well with $P_{data}(\mathbf{x}_t|\mathbf{x}_{1:t-1})$; or 2) the computational constraints limiting the number of sampled queries, leading to inaccurate estimates.

**Aggregation**    Aggregation based on answers and verbalized confidences (e.g., Avg-Conf and Pair-Rank) outperforms aggregation based on answers only (e.g., consistency), especially when LLM queries are costly and the number of queries we can sample is constrained. This is due to the coarse granularity of the consistency-based aggregation's output—limited to 6 possible values (0, 0.2, 0.4, 0.6, 0.8, 1) when M=5. This can lead to poor calibration performance. The verbalized confidence, despite being less precise, still captures the model's uncertainty tendency and allows for finer-grained output values, and hence can be combined to enhance calibration performance.

**Verbalized Confidence**    For verbalized confidence, we note that humans are able to verbalize their uncertainty, e.g., giving insight as to whether our answers and reasonings are correct or not. So it is reasonable to expect LLMs to have also learned this ability, or to learn it at some point in the future. The current suboptimal performance of verbalized confidence points to an important research gap, and this might be explained by the inherent inaccuracy of the training data, particularly human expressions of uncertainty. For example, as studied by Garthwaite et al. (2005a), humans sometimes tend to exaggerate their a priori probability for an event that has occurred.

**Prompting Strategy**    In addition, compared to Vanilla prompt, Top-K, CoT, and Multi-Step can significantly reduce ECE in ChatGPT. We argue that the improvement is largely due to these prompt strategies enhancing the model's accuracy, which narrows the gap between average confidence and actual accuracy, rather than a significant boost in their ability to differentiate between correct and incorrect samples. This is also supported by the modest gains in AUROC and AUPRC, compared to the significant improvement in ECE.

## E   EXPERIMENT SETUP

### E.1   DATASETS

To evaluate the quality of confidence estimates in varied tasks, we select the tasks of commonsense reasoning, arithmetic calculation, symbolic reasoning, professional knowledge, and ethical knowledge as evaluation benchmarks. In detail, the datasets for each task are listed below:

- **Commonsense Reasoning**: Sports Understanding (SportUND) dataset (Kim, 2021) and StrategyQA dataset (Geva et al., 2021) from BigBench (Ghazal et al., 2013). We select StrategyQA as the more representative dataset since it contains more data.

- **Arithmetic Reasoning**: Graduate School Math (GSM8K) dataset (Cobbe et al., 2021) and Simple Variations on Arithmetic Math word Problems (SVAMP) dataset (Patel et al., 2021). We select GSM9K as the more representative dataset because it has a wider usage.

- **Symbolic Reasoning**: Date Understanding (DateUnd) dataset (Wu & Wang, 2021) and Object Counting (ObjectCou) dataset (Wang et al., 2019) in BigBench. We select Date Understanding as the more representative dataset since it is more difficult than Object Counting.

- **Professional Knowledge**: Professional Law (Prf-Law) dataset from MMLU (Massive Multitask Language Understanding)  (Hendrycks et al., 2021)

- **Ethical Knowledge**: business ethics (Biz-Ethics) dataset from MMLU (Hendrycks et al., 2021).

### E.2 EVALUATION METRICS

In line with previous evaluation setting in (Naeini et al., 2015; Yuan et al., 2021; Xiong et al., 2022), we use confidence calibration and failure prediction metrics to measure estimated confidence:

- Expected Calibration Error (**ECE**): It measures the calibration of a classifier by quantifying the discrepancy between predicted probabilities and observed accuracy.
- Area Under the Receiver Operating Characteristic curve (**AUROC**): It assesses the discriminative ability of a classifier across different classification thresholds (Boyd et al., 2013).
- Area under the Precision-Recall Curve (**AUPRC**): It measures the trade-off between precision and recall at different classification thresholds. Specifically, AUPRC-Positive measures the AUPRC for positive instances and AUPRC-Negative is for negative samples.

Specifically, calibration metrics (ECE) measure the alignment of confidence scores with the ground truth uncertainty, enabling their utilization in tasks such as risk assessment; while failure detection (AUROC and AUPOR) metrics measure whether the confidence score can appropriately differentiate correct answers and incorrect answers. These metrics also play a crucial role in accurately assessing calibration measurements in works such as Mielke et al. (2022) and Solano et al. (2021).

### E.3 MODELS

In our experiments, we incorporate a range of representative LLMs of different scales, including Vicuna (Chiang et al., 2023), GPT3 (Brown et al., 2020), GPT3.5 (GPT3.5) (OpenAI, 2021), and GPT4 (OpenAI, 2023). The number of parameters in each model is 13 billion for Vicuna, 175 billion for GPT3, and larger for GPT3.5 and GPT4. While GPT3.5 and GPT4 have been widely acknowledged due to their outstanding performances, GPT3 is selected as a former version of them. Vicuna is a smaller model fine-tuned from LLaMA (Touvron et al., 2023a).

> ## Example: Open-Number Question
>
> Read the question, provide your answer and your confidence in this answer. Note: The confidence indicates how likely you think your answer is true.
>
> Use the following format to answer:
>
> ```Answer and Confidence (0-100): [ONLY the number; not a complete sentence], [Your confidence level, please only include the numerical number in the range of 0-100]\%```
>
> Only the answer and confidence, don't give me the explanation
>
> Question: A robe takes 2 bolts of blue fiber and half that much white fiber. How many bolts in total does it take?
>
> Now, please answer this question and provide your confidence level.
>
> ------------------------
>
> Answer and Confidence: 3, 85%

Figure 8: Example of a complete prompt and the model's output. The vanilla prompt is used.

### E.4 IMPLEMENTATION DETAILS

For the use of sampling strategy, we sample $M = 5$ responses. For the use of Self-Random, we set the temperature hyper-parameter as 0.7 to gather a more diverse answer set, as suggested in Wang et al. (2022). The p

# F  PROMPTS

The prompts used in our work consist of three components: the description, the question, and the misleading hints (used for misleading sampling strategy). The description part outlines the definition of the task presented to the LLMs, requesting them to provide an answer together with the confidence level for the answer. See Figure 8 for a complete example of full prompt and the model's output. The detailed prompt is provided below:

1. Vanilla: Table 14
2. Chain-of-Thought-based: Table 15
3. Self-Probing: Table 16
4. Multi-Step: Table 17
5. Top-K: Table 18

Table 14: The designed vanilla prompt for two different tasks.

| Task | Vanilla Prompt |
|---|---|
| Multi-choice questions | Read the question, provide your answer and your confidence in this answer. Note: The confidence indicates how likely you think your answer is true.

Use the following format to answer:

"'Answer and Confidence (0-100): [ONLY the option letter; not a complete sentence], [Your confidence level, please only include the numerical number in the range of 0-100]%"'
Only the answer and confidence, don't give me the explanation.

Question:[Specific Question Here]

Now, please answer this question and provide your confidence level. |
| Open-number questions | Read the question, provide your answer and your confidence in this answer. Note: The confidence indicates how likely you think your answer is true.

Use the following format to answer:

"'Answer and Confidence (0-100): [ONLY the number; not a complete sentence], [Your confidence level, please only include the numerical number in the range of 0-100]%"'

Only the answer and confidence, don't give me the explanation.

Question:[Specific Question Here]

Now, please answer this question and provide your confidence level. |

Table 15: The prompt designed for Chain-of-Thought prompting strategy.

| Tasks | Definitions of Tasks in Prompts in Chain-of-Thought Confidence |
|---|---|
| Multi-choice questions | Read the question, analyze step by step, provide your answer and your confidence in this answer. Note: The confidence indicates how likely you think your answer is true.

Use the following format to answer:

"'Explanation: [insert step-by-step analysis here]
Answer and Confidence (0-100): [ONLY the option letter; not a complete sentence], [Your confidence level, please only include the numerical number in the range of 0-100]%"'

Only give me the reply according to this format, don't give me any other words.

Question:[Specific Question Here]

Now, please answer this question and provide your confidence level. Let's think it step by step. |
| Open-number questions | Read the question, analyze step by step, provide your answer and your confidence in this answer. Note: The confidence indicates how likely you think your answer is true.

Use the following format to answer:

"'Explanation: [insert step-by-step analysis here]
Answer and Confidence (0-100): [ONLY the number; not a complete sentence], [Your confidence level, please only include the numerical number in the range of 0-100]%"'

Only give me the reply according to this format, don't give me any other words.

Question:[Specific Question Here]

Now, please answer this question and provide your confidence level. Let's think it step by step. |

Table 16: The prompt designed for self-probing prompting strategy.

| The prompt designed for self-probing prompting strategy |
|---|
| Question: [The specific question]

Possible Answer: [The answer candidates]

Q: How likely is the above answer to be correct? Please first show your reasoning concisely and then answer with the following format:

"'Confidence: [the probability of answer {answer} to be correct, not the one you think correct, please only include the numerical number]"' |

Table 17: The designed prompt for multi-step prompting strategy.

| | The designed prompt for multi-step prompting strategy |
|---|---|
| Question | Read the question, break down the problem into K steps, think step by step, give your confidence in each step, and then derive your final answer and your confidence in this answer. Note: The confidence indicates how likely you think your answer is true.

Use the following format to answer:
"'Step 1: [Your reasoning], Confidence: [ONLY the confidence value that this step is correct]%
...
Step K: [Your reasoning], Confidence: [ONLY the confidence value that this step is correct]%
Final Answer and Overall Confidence (0-100): [ONLY the answer type; not a complete sentence], [Your confidence value]%'" |

Table 18: Prompts used to elicit Top-K Verbalized Confidence.

| | The designed prompt for Top-K prompting strategy. |
|---|---|
| Question | Provide your k best guesses and the probability that each is correct (0% to 100%) for the following question. Give ONLY the task output description of your guesses and probabilities, no other words or explanation. For example:
G1: <ONLY the task output description of first most likely guess; not a complete sentence, just the guess!> P1: <ONLY the probability that G1 is correct, without any extra commentary whatsoever; just the probability!>

...
Gk: <ONLY the task output description of k-th most likely guess> Pk: <ONLY the probability that Gk is correct, without any extra commentary whatsoever; just the probability!> |

