# OpenReview forum: "Can LLMs Express Their Uncertainty? An Empirical Evaluation of Confidence Elicitation in LLMs"
_ICLR.cc/2024/Conference — ICLR 2024 poster_

### Official Review · Reviewer_bmdf · 2023-10-30

**Soundness:** 3 good
**Presentation:** 3 good
**Contribution:** 2 fair
**Rating:** 5
**Confidence:** 3

**Summary:**

The paper evaluateds an LLM's capability to express uncertainty. To this end they define a systematic evaluation framework.

+ LLMs are a highly relevant topic
+ Elicitating confidence is also an important topic in deep learning in general
+ The study performs extensive evaluation on multiple datasets and LLMs
- The evaluation and outcomes yield little novel insights (see also detailed comments)
- The methods are standard, a methodological contribution would be highly appreciated. The idea to use LLMs for self-assessment is all-to-common. That said, it is also interest to assess it in this context, but on its own a marginal contribution. The prompting/aggregation techniques at best partially mitigate this, but they are also well-known. In short, more would be expected for a strong contribution.


Detailed comments?
> 1) LLMs, when verbalizing their confidence, tend to be overconfident...
LLMs are known to be "Fluent but non-factual" or "Convincing though wrong". Given that this study focuses only on evaluation I would have expected more novel findings as key fining.
> 2) As model capability scales up, both calibration and failure prediction performance improve, yet still far from ideal performance.
That larger LLMs are better (on all benchmarks) than smaller ones is also general knowledge - check the LLama2 paper, for example, and compare benchmarks for 7B,13B...
> It would be interesting (and maybe even necessary) to study how much the uncertainty estimates depends on the prompt, i.e., if you state in the prompt "Appear confident" are estimates more overconfident?

**Strengths:**

see above

**Weaknesses:**

see above

**Questions:**

see above

---

> ### Author Response · Authors · 2023-11-17
> **Clarification of our findings' contribution & Impact of the prompt**
>
> We appreciate reviewer bmdf's constructive feedback. We are glad the reviewer finds the topic important, and the evaluation systematic and extensive. Here we answer all the questions and hope they can address the concerns.
>
> ----
>
> ### **C1: LLMs are known to be "Fluent but non-factual" or "Convincing though wrong". Given that this study focuses only on evaluation I would have expected more novel findings.**
>
> Firstly, we concur with your observation that LLMs can be "Fluent but non-factual" or "Convincing though wrong". However, we would like to clarify that uncertainty and factuality, despite their similarity in concept, measure the model's two different aspects–a model can exhibit both non-factual outputs and provide well-calibrated uncertainty estimates. While the aspect of factuality (e.g., hallucination) has been extensively studied, the uncertainty in LLMs remains underexplored. This leads to the critical, yet unresolved question: can LLMs accurately express uncertainty? Our work is indeed motivated from here, and we aim to delve further into: 1) Can we utilize better prompt strategies to elicit confidence accurately? 2) What alternative methods could better estimate uncertainty? How is their performance? 3) Can we recommend some stable combination algorithms for practitioners that can generalize?
>
> To answer these questions, we have first proposed an evaluation framework to analyze which prompting, sampling, and aggregation strategies can effectively mitigate overconfidence. Following the reviewer's suggestions, we have included a discussion in Sec 6 of our paper (highlighted in blue text)  that offers recommendations for practitioners to elicit the model’s confidence, which we believe are valuable contributions from our empirical analysis. We kindly refer the reviewer to the paper for more information.
>
> While some of our findings may not seem surprising, we believe our research, as the first research work to extensively study this problem, can provide a foundation for further study and call for more attention to the uncertainty of the model's prediction in addition to factuality.
>
>
> ----
>
> ### **C2: Larger LLMs are better (on all benchmarks) than smaller ones is also general knowledge.**
>
> Thank you for the valuable feedback. We agree with you that larger LLMs perform better on a wide variety of tasks. However, for the task of calibration, it has been found that larger models are not always well-calibrated: for example, GPT4 fine-tuned on RLHF indeed hurts the calibration performance[1]. Furthermore, in a black-box setting, particularly when the model is not trained to output verbalized confidence, it still requires experiments to confirm whether this common knowledge can be generalized.
>
> [1] OpenAI, R. (2023). GPT-4 technical report. arXiv, 2303-08774.
>
> ----
>
> ### **C3: It would be interesting (and maybe even necessary) to study how much the uncertainty estimates depends on the prompt, i.e., if you state in the prompt "Appear confident" are estimates more overconfident?**
>
> Thank you for the interesting idea! Inspired by the reviewer's suggestion, we have explored how prompts like **"You are a confident GPT"** and **"You are a cautious GPT"** affect confidence elicitation performance and found small differences in accuracy and uncertainty metrics, as shown below. In addition, we also visualize the plots of the output confidence distribution and find that they are quite similar. This suggests that this type of variation in prompt has a limited impact on the performance compared to the prompting strategies studied in Sec 3.2. The visualization plots, implementation details and more analysis can be found in Appendix Sec B.1.
>
> | Role       | Dataset | Model        | ACC     | ECE     | AUROC   | AUPRC_P | AUPRC_N |
> |------------|---------|--------------|---------|---------|---------|---------|---------|
> | Confident  | GSM8K   | chatgpt-0613 | 0.7103 | 0.2741 | 0.5679 | 0.7398 | 0.3635 |
> | Cautious   | GSM8K   | chatgpt-0613 | 0.6983  | 0.2812 | 0.5946 | 0.7415 | 0.4009 |
>
> Additionally, we would like to share an interesting experiment we previously conducted for the misleading sampling strategy in the original submission (see Appendix Sec B.5). We append hint texts with varying degrees of certainty (Table 8) to the prompts, such as **strong claim**: "I am 100% confident that the answer should be xx", **weak claim**:  "I vaguely remember the answer is" or the **external source**: "The Wikipedia says the answer should be xx”. Then we measure the uncertainty by computing how likely the model will change their original prediction. Interestingly, our experiments showed that prompts with weaker confidence descriptions (weak claims) often yielded better results than those with strong claims or external course hints. Details are provided in our paper's appendix Sec B.5.

---

> > ### Comment · Reviewer_bmdf · 2023-11-21
> >
> > I have read the author response.

---

> > > ### Author Response · Authors · 2023-11-22
> > >
> > > Dear Reviewer Bmdf,
> > >
> > > Thank you for taking the time to read our response. We have noticed that you have given a negative rating, and we are concerned that there may be issues that remain unaddressed. Please let us know if there are any other matters or concerns, and what would be expected to improve your rating. We are committed to doing our best to address them.
> > >
> > > Best regards,
> > > The Authors

---

### Official Review · Reviewer_uQRx · 2023-11-01

**Soundness:** 4 excellent
**Presentation:** 4 excellent
**Contribution:** 3 good
**Rating:** 8
**Confidence:** 4

**Summary:**

The paper proposes a unified framework under black box setting to evaluate how calibrated large language models are for their predictions. Specifically, no logits or internal states of LLM are assumed given but the models can be asked to provide explicit confidence. The authors evaluate 5 different prompting strategy and various aggregation strategy to elicit better calibrated results from the models.

**Strengths:**

1. The paper is very well-written and tackles the black box confidence calibration in a timely manner. The approach is very targeted to LLMs which provide many insights that are not available from calibrating image classification models.

2. The evaluation is very thorough and the two calibration tasks defined are reasonable.

**Weaknesses:**

1. It seems black-box based confidence elicitation approaches are generally unsatisfactory in calibration performance. Can the authors directly compare them with white-box based calibration approaches such as token probability based calibration approach on open-source models and observe how large the gap can be on ECE and failure prediction?

2. Equation 3. Shouldn't we maximize with respect to P since we are doing an MLE estimation?

**Questions:**

Please see the weakness section above.

---

> ### Author Response · Authors · 2023-11-17
> **The comparison of white-box methods and black-box methods will be updated soon.**
>
> We sincerely thank reviewer uQRx for the constructive suggestion on the white-box comparison experiment. We are glad that the reviewer finds the paper well-written, the finding meaningful, and the evaluation thorough. Here we first clarify the questions and update later with experiment results to address the concerns.
>
> ----
>
>
> ### **W1:Compare with white-box based calibration approaches such as token probability-based calibration approach on open-source models and observe how large the gap can be on ECE and failure prediction?**
>
> Thanks for this constructive suggestion! We are conducting experiments comparing black-box and white-box calibration approaches now and will update the results soon.
>
>
> ### **Q2: Equation 3. Shouldn't we maximize with respect to P since we are doing an MLE estimation?**
>
> Thank you for pointing out the discrepancy in Equation 3. You're correct; it should maximize with respect to P. This has been corrected in the revised manuscript.

---

> > ### Author Response · Authors · 2023-11-22
> > **Comparative Analysis of White-Box and Black-Box Methods**
> >
> > We sincerely appreciate this suggestion! To address these questions, we conduct a comparative analysis of token-probability-based **white-box** methods against **black-box** models utilizing verbalized confidence. Below we highlight some of the important results and details. More results and details can be found in Appendix B.1 of the revised manuscript.
> >
> > ----
> >
> > **Implementation details**:
> > We utilize the probabilities of each output token to develop three token-probability-based white-box methods:
> > 1. **Sequence Probability (seq-prob)**, which aggregates the probabilities of all tokens.
> > 2. **Length-Normalized Sequence Probability (len-norm-prob)**, which normalizes the sequence probability based on sequence length, i.e., $\text{seq-prob}^{\text{1/length}}$
> > 3. **Key Token Probability (token-prob)**, designed to focus on the result-specific tokens, e.g., **"35"** in the output sequence *"Explanation: ....; Answer: 35; ..."*, thereby minimizing the influence of irrelevant output tokens.
> >
> > For our implementation, we use the best-performing Top-K and the commonly used CoT (Chain-of-Thought) as the prompting strategies to obtain the verbalized confidence and select GPT3 as the backbone model. The table below shows the performance using Top-K as the prompting strategy. The metrics are given by $\times 10^2$.
> >
> > ---
> >
> > | Dataset   | Acc   | Method           | ECE   | AUROC | AUPRC-P | AUPRC-N |
> > |-----------|-------|------------------|-------|-------|---------|---------|
> > | StrategyQA| 59.90 | Verbalized       | 39.04 | 50.34 | 60.06   | 40.27   |
> > |           |       | seq-prob         | **7.14**  | 55.50 | 62.99   | 45.22   |
> > |           |       | len-norm-prob    | 37.65 | 55.50 | 62.99   | 45.22   |
> > |           |       | token-prob       | 32.43 | **60.61** | **69.90**  | **47.10**  |
> > | Biz-Ethics| 61.00 | Verbalized       | **18.20** | 66.27 | 71.95   | 50.59   |
> > |           |       | seq-prob         | 48.49 | 62.30 | 71.07   | 52.23   |
> > |           |       | len-norm-prob    | 33.70 | 62.30 | 71.07   | 52.23   |
> > |           |       | token-prob       | 27.65 | **67.00** | **74.89**  | **55.01**  |
> > | GSM8K     | 11.52 | Verbalized       | 77.40 | 54.05 | 12.70   | 89.01   |
> > |           |       | seq-prob         | **7.73**  | 69.80 | 20.40   | 94.71   |
> > |           |       | len-norm-prob    | 72.41 | **70.61** | **21.23**  | **94.75**  |
> > |           |       | token-prob       | 35.60 | 69.29 | 20.63   | 94.27   |
> > | DateUND   | 15.72 | Verbalized       | 83.47 | 50.80 | 15.93   | 84.54   |
> > |           |       | seq-prob         | **16.10** | **62.93** | **22.39**  | **90.61**  |
> > |           |       | len-norm-prob    | 81.27 | **62.93** | **22.39** | **90.61**  |
> > |           |       | token-prob       | 74.19 | 54.25 | 19.28   | 83.85   |
> > | Prf-Law   | 44.92 | Verbalized       | 41.55 | 49.54 | 44.43   | 55.78   |
> > |           |       | seq-prob         | **32.31** | 51.07 | 45.75   | 56.70   |
> > |           |       | len-norm-prob    | 49.66 | 51.06 | 45.75   | 56.79   |
> > |           |       | token-prob       | 43.26 | **61.24** | **53.84** | **64.69** |
> >
> > ----
> >
> > **Findings**:
> > Our comparative analysis, detailed in Table 5 and Table 6 of the paper (we also show the Table 6 below), yields several key insights:
> > 1. Generally, **white-box methods exhibit better performance**, with length-normalized sequence probability and key token probability emerging as the most effective methods across five datasets and four evaluation metrics.
> > 2. **The gap between white-box and black-box methods is relatively modest**. Moreover, even the best-performing **white-box methods fall short of achieving satisfactory results**. This is particularly apparent in the AUROC metric, where the performance of nearly all methods across various datasets ranges between 0.5-0.6, signifying a limited capability in distinguishing between correct and incorrect responses.
> > 3. These experimental results suggest that **uncertainty estimation in LLMs remains a challenging and unresolved issue**. As mentioned in our introduction, the logit-based methods, which predominantly capture the model's uncertainty regarding the next token, are less effective in capturing the semantic uncertainty inherent in their textual meanings[1]. Although several alternative approaches like semantic uncertainty [1] have been proposed, they come with significant computational demands. This scenario underscores the need for future research on both white-box and black-box methods to discover more efficient and effective methods for uncertainty estimation in LLMs.
> >
> > [1] Kuhn, L., Gal, Y., & Farquhar, S. (2023). Semantic uncertainty: Linguistic invariances for uncertainty estimation in natural language generation. ICLR 2023.

---

### Official Review · Reviewer_Bvbz · 2023-11-02

**Soundness:** 4 excellent
**Presentation:** 3 good
**Contribution:** 3 good
**Rating:** 6
**Confidence:** 3

**Summary:**

A comprehensive study on self-estimation of uncertainty by LLMs. The authors introduce multiple design choices that can affect uncertainty elicitation and then evaluate them on existing LLMs.

**Strengths:**

- Uncertainty estimation is an important question for LLMs. The findings of this study can be impactful in practice
- The 3-part framework of prompting strategy, sampling and aggregation provide clarity.
-The evaluation is comprehensive across many datasets.

**Weaknesses:**

The strategies presented are reasonable (but not super novel, as far as I understand). So the contribution should be evaluated on the empirical evaluation. Here, it is unclear how to interpret the results. No strategy seems to be work well always.

I understand it is hard to analyze black-box LLMs, but still some discussion on why some strategy works, and why it does not, will be useful.

What is the best combination of prompting, sampling and aggregation? can the authors propose the best way for practitioners while also noting the limitations? Right now, the message seems to be that nothing works. But it may be useful to present a version of the best performing method.

**Questions:**

see the weaknesses above.

---

> ### Author Response · Authors · 2023-11-17
> **Discussions on why some strategies work, and why some do not work.**
>
> We sincerely thank reviewer Bvbz for the constructive suggestion on the discussion of why some strategies work, and why some do not. We are glad that the reviewer finds the question important, the study comprehensive, the paper with clarity, and the findings impactful in practice. Here we answer all the questions and hope they can address the concerns.
>
> -----
>
> ### **Q1: Discussions on why some strategies work, and why some do not work.**
>
> Thank you for this constructive suggestion. We have added a dedicated discussion to Appendix C and we highlight the discussion of some effective strategies and advice here:
>
> 1. *Consistency among multiple responses is more effective compared to verbalized confidence ($M=1$), with particularly notable improvements on the arithmetic task* (Sec 5.3).
>     - This is because sampling more queries allows us to directly approximate the model's internal distribution, $P\_{model}(\mathbf{x}\_t|\mathbf{x}\_{1:t-1})$, which is trained to mirror the ground truth data distribution.
>     - Issues making this method ineffective can be: 1) the model's poor calibration [1], i.e., $P\_{model}(\mathbf{x}\_t|\mathbf{x}\_{1:t-1})$ does not align well with $P\_{data}(\mathbf{x}\_t|\mathbf{x}\_{1:t-1})$; or 2) the computational constraints limiting the number of sampled queries, leading to inaccurate estimates.
>
> 2. *Aggregation based on answers and verbalized confidences* (e.g., Avg-Conf and Pair-Rank) outperforms *aggregation based on answers only (e.g., consistency)* (Sec 5.4).
>     - This advantage is particularly notable when LLM queries are costly and the number of queries we can sample is constrained.
>     - This is due to the coarse granularity of the consistency-based aggregation's output—limited to 6 possible values (0, ⅕, ⅖, ⅗, ⅘, 1) when M=5. This can lead to poor calibration performance. The verbalized confidence, despite being less precise, still captures the model's uncertainty tendency and allows for finer-grained output values, and hence can be combined to enhance calibration performance (see Table 4).
>
> 3. For verbalized confidence, we note that humans are able to verbalize their uncertainty, e.g., giving insight as to whether our answers and reasonings are correct or not. So it is reasonable to expect LLMs to have also learned this ability, or to learn it at some point in the future.
>    - The current suboptimal performance of verbalized confidence points to an important research gap, and this might be explained by the inherent inaccuracy of the training data, particularly human expressions of uncertainty. For example, as studied by [2], humans sometimes tend to exaggerate their a priori probability for an event that has occurred.
>
> 3. Compared to Vanilla prompt, Top-K, CoT, and Multi-Step can significantly reduce ECE in ChatGPT.
>      - We argue that the improvement is largely due to these prompt strategies enhancing the model's accuracy, which narrows the gap between average confidence and actual accuracy, rather than a significant boost in their ability to differentiate between correct and incorrect samples. This is also supported by the modest gains in AUROC and AUPRC, compared to the significant improvement in ECE.
>
> [1] Kuhn, Lorenz, Yarin Gal, and Sebastian Farquhar. "Semantic uncertainty: Linguistic invariances for uncertainty estimation in natural language generation." ICLR (2023).
>
> [2] Garthwaite, Paul H., Joseph B. Kadane, and Anthony O'Hagan. "Statistical methods for eliciting probability distributions." Journal of the American statistical Association (2005)

---

> ### Author Response · Authors · 2023-11-17
> **Propose the best way for practitioners while also noting the limitations**
>
> ### **Q: Can the authors propose the best way for practitioners while also noting the limitations?**
>
> Thanks for the constructive suggestions. We have added a discussion and conclusion on the best way for practitioners into Section 6 of the paper. Specifically, we recommend the practitioners use “Top-K prompt + Self-Random sampling (M=5) + Avg-Conf/Pair-Rank aggregation” as the best method so far due to their stable performances.
>
> The recommendation is made by balancing efficiency, simplicity, and effectiveness through multiple benchmark datasets:
> - Top-K outperforms all other methods on GPT-3.5 and is comparable to the top-performing method Self-Probing on GPT4. Compared to Self-Probing which requires two inference phases, the Top-K prompt is chosen for the balance between effectiveness and efficiency.
> - As shown in Sec 5.3, ensemble methods (e.g., $M=5$) are consistently more effective than verbalized confidence ($M=1$) in eliciting a model's confidence. Regarding the sampling strategies, Self-Random is selected for being more straightforward and commonly used, since the performance difference of different sampling strategies is minimal.
> - For aggregation, strategies based on both answers and verbalized confidences (e.g., Avg-Conf and Pair-Rank) outperform *aggregation based on answers only (e.g., consistency)*. Then we recommend Pair-Rank and Avg-Conf for different downstream tasks according to their relatively good performance on different metrics. For example, for tasks that prioritize the exact confidence values, like calculating expected risk, Pair-Rank is recommended, while Avg-Conf is better suited for tasks related to failure prediction, e.g., factual error detection.

---

### Official Review · Reviewer_GBf5 · 2023-11-03

**Soundness:** 2 fair
**Presentation:** 3 good
**Contribution:** 2 fair
**Rating:** 5
**Confidence:** 4

**Summary:**

This paper studies approaches to eliciting reliable confidence estimates from large language models about their statements.  The paper studies closed-box methods for confidence elicitation exploring (1) prompting strategies for verbalized confidence; (2) sampling approaches to measure variance across multiple responses; and (3) aggregation methods to compute consistency measures.

**Strengths:**

- This paper studies an important problem.  When an LLM's reliability is critical to an application or higher-level task, knowing the likelihood that the LLMs responses is correct or not is critical.

The overall approach, combining prompting, sampling, and aggregation, is an elegant approach for closed-box confidence solicitation, if it can be calibrated or its reliability as a measure otherwise completely characterized.

The experiments span many task-domains and multiple models.

**Weaknesses:**

The experimental results show that the method is useful in identifying uncertainty, but performance varies significantly across benchmark tasks and no single method is clearly better than others.  Given a new task, it is not clear which method will give the best performance, or even if the best performing method on a hold out set will generalize to real problems in a domain.  I appreciate that the authors call this out in the discussion, saying that none of the current algorithms are satisfactory.

The high level observations enumerated in the introduction are interesting but also seem very preliminary, and difficult to operationalize or build upon.



minor:
- the acronym ECE should be defined at first use.
- typo page 6: "aevaluation" -> "evaluation"

**Questions:**

Is there a reason to believe that the model's certainty _should_ be correlated with model correctness?  Or to understand a priori (e.g., based on training data) when it might be more or less likely to be so?

Minor variations in prompt wording can have significant effects on performance.  How was the specific prompt text chosen for each of the prompt strategies?  was it optimized or experimented with?

---

> ### Author Response · Authors · 2023-11-17
> **The correlation between model certainty and correctness & How is the prompt chosen**
>
> ### **Q1: Is there a reason to believe that the model's certainty should be correlated with model correctness? Or to understand a priori (e.g., based on training data) when it might be more or less likely to be so?**
>
> This is indeed an excellent question! In fact, the correlation depends on the definition of uncertainty, e.g., data/model uncertainty[1]. Here to answer the reviewer’s question, we clarify the following subquestions:
>
> 1. What is our definition of model certainty?
> 2. Does this defined model certainty reflect model correctness?
>
> In line with established research [1,2,3,4,6], we define the 'ground-truth' certainty of a model's prediction $\mathbf{x}\_t$ as the corresponding probability from the data distribution $P_{data}(\mathbf{x}\_t|\mathbf{x}\_{1:t-1})$. This probability is highly correlated with model correctness, that is, a lower probability implies that the prediction is less likely to be correct. Therefore, we believe that **the model’s certainty should be correlated with model correctness**, which is also practically useful, e.g., help humans identify potential factual errors of LLMs.
>
> 3. Can our proposed uncertainty scores capture the model correctness?
>
> We suspect the reviewer's question might also arise from doubts about whether our proposed consistency or verbalized confidence can capture model correctness, e.g., the models can be confident but wrong. **We would like to clarify that our methods are designed with the potential to capture correctness**, although issues such as computational constraints can affect their efficacy:
>
> - For consistency, sampling more queries theoretically allows us to approximate the model's internal distribution, thus capturing correctness. The issues making this method ineffective can be the model's poor calibration [4] or the computational constraints. For example, the limited number of sampled queries might lead to inaccurate estimates.
>
> - For verbalized confidence, we note that humans are able to verbalize their uncertainty, e.g., giving insight as to whether our answers and reasonings are correct or not. So it is reasonable to expect LLMs to have also learned this ability, or to learn it at some point in the future. LLM’s current inaccuracy at this task points to an important research gap and might be explained by the training data being affected by the observed overconfident tendency of human uncertainty expression[5].
>
> ----
>
> - [1] Malinin, Andrey, and Mark Gales. "Predictive uncertainty estimation via prior networks." NeurIPS (2018).
> - [2] Guo, Chuan, Geoff Pleiss, Yu Sun, and Kilian Q. Weinberger. "On calibration of modern neural networks." ICML 2017.
> - [3] Liu, Jeremiah, et al. "Simple and principled uncertainty estimation with deterministic deep learning via distance awareness." NeurIPS (2020)
> - [4] Kuhn, Lorenz, Yarin Gal, and Sebastian Farquhar. "Semantic uncertainty: Linguistic invariances for uncertainty estimation in natural language generation." ICLR (2023).
> - [5] Garthwaite, Paul H., Joseph B. Kadane, and Anthony O'Hagan. "Statistical methods for eliciting probability distributions." Journal of the American statistical Association (2005)
> - [6] Jiang, Zhengbao, et al. "How can we know when language models know? on the calibration of language models for question answering." ACL (2021).
>
> ----
>
> ### **Q2: How was the specific prompt text chosen for each of the prompt strategies? was it optimized or experimented with?**
>
> Thanks for the question. Indeed, the prompts (see Appendix Sec D for all the used prompts) are selected through iterative refinement to avoid potential ambiguity and bias, and we are happy to share some insights from this refinement process.
>
> Initially, we started with a straightforward prompt like "Provide your confidence in this answer." However, we quickly discovered that the model's responses varied widely, ranging from textual descriptors like "confidence: high," to statements such as "I am very confident," and even disclaimers like "as AI language models, I cannot provide confidence."
>
> In addition, we observed that providing examples in the prompts can insert bias into the distribution of outputs. For instance, saying "ONLY the option letter, e.g., C" would inadvertently increase the frequency of 'C' as a selected answer in weaker models like Vicuna.
>
> Consequently, we refined our prompts further by removing such examples to reduce bias. These improvements were made step by step, informed by careful observation and analysis of the output results. While we cannot guarantee that the current prompts are the optimal versions, they have been designed to minimize the interference of irrelevant factors as much as possible.

---

> > ### Comment · Reviewer_GBf5 · 2023-11-22
> >
> > Thank you.  I appreciate the authors' answers to my questions.

---

> > > ### Author Response · Authors · 2023-11-22
> > >
> > > Dear Reviewer GBf5,
> > >
> > > Thank you for taking the time to read our response. We have noticed that you have given a negative rating, and we are concerned that there may be issues that remain unaddressed. Please let us know if there are any other matters or concerns and what would be expected to improve your rating. We are committed to doing our best to address them.
> > >
> > > Best regards,
> > >
> > > The Authors

---

> ### Author Response · Authors · 2023-11-17
> **Recommendations for the practitioners to operationalize or build upon**
>
> We thank the reviewer Gbf5 for recognizing the importance of our work, the elegance of our approach, and the appreciation for our discussion. Here we answer all the questions and hope they can address the concerns.
>
> -----
>
> ### **Weakness: The high-level observations are interesting but difficult to operationalize or build upon.**
>
> Thanks for the helpful feedback. To guide practitioners on how to utilize our findings, we have revised the paper to include a dedicated discussion in Section 6.
>
> **TL;DR: we highlight simple yet effective strategies and recommend the practitioners use “Top-K prompt + Self-Random sampling + Avg-Conf/Pair-Rank aggregation” as the best method so far due to their stable performance.**
>
> The recommendation is made by balancing efficiency, simplicity, and effectiveness through multiple benchmark datasets:
> - Top-K outperforms all other methods on GPT-3.5 and is comparable to the top-performing method Self-Probing on GPT4. Compared to Self-Probing which requires two inference phases, the Top-K prompt is chosen for the balance between effectiveness and efficiency.
> - As shown in Sec 5.3, ensemble methods (e.g., $M=5$) are consistently more effective than verbalized confidence ($M=1$) in eliciting a model's confidence. Regarding the sampling strategies, Self-Random is selected for being more straightforward and commonly used, since the performance difference of different sampling strategies is minimal.
> - For aggregation, strategies based on both answers and verbalized confidences (e.g., Avg-Conf and Pair-Rank) outperform *aggregation based on answers only (e.g., consistency)*. Then we recommend Pair-Rank and Avg-Conf for different downstream tasks according to their relatively good performance on different metrics. For example, for tasks that prioritize the exact confidence values, like calculating expected risk, Pair-Rank is recommended, while Avg-Conf is better suited for tasks related to failure prediction, e.g., factual error detection.

---

### Author Response · Authors · 2023-11-17
**Summary of major changes & Highlight of the suggestions for practitioners**

We sincerely thank all the reviewers for their helpful comments and suggestions. We now have uploaded the rebuttal version of our paper together with the appendix where the revisions are marked in **blue**.

Here is the summary of the major changes we made in the revision:
- Additional white-box vs black-box comparison results in Appendix B.1 (Table 5, 6)
- Additional prompt results in Appendix B.2 (Fig. 4, Table 7);
- A new Appendix D with practical suggestions and considerations to practitioners
- A new Appendix E with the discussion of the working mechanism of the strategies
- Sec 5 and 6 have been updated with more detailed discussions to improve clarity.
- We have fixed some minor issues such as typos.


Additionally, we highlight some of the practical guidance beyond LLMs' overconfidence for practitioners to use and further improve the framework:
1. Prioritize ensemble-based methods compared to verbalized confidence.
2. When aggregating responses from ensemble-based methods, incorporating both verbalized confidence and answer consistency yields better results than considering consistency alone.
3. We recommend the practitioners use “Top-K prompt + Self-Random sampling + Avg-Conf/Pair-Rank aggregation” as the best method so far to use due to their stable performances. (see rationale in Sec 6 and Appendix D of the manuscript).

---

> ### Author Response · Authors · 2023-11-22
> **Adding Comparative Analysis of White-box vs Black-box Methods To Appendix B.1**
>
> In the revised manuscript, we have incorporated a comparative analysis between black-box methods and their white-box counterparts, e.g., sequence probability. For more results and details, please see the response to reviewer uQRx at (https://openreview.net/forum?id=gjeQKFxFpZ&noteId=h8O7QAxL4y) and Appendix B.1 in the paper.

---

> > ### Author Response · Authors · 2023-11-23
> > **General Response**
> >
> > We would like to thank all the reviewers for their constructive feedback and suggestions, which have been instrumental in improving our paper.
> >
> > ---
> >
> > Below is a summary of the major concerns raised and how we have addressed them:
> >
> > - **Lack of practical suggestions**: we have revised our manuscript to present a version of the stable-performing method. We also discuss why certain strategies work and others do not, and we have added principles and considerations on how employ black-box methods effectively (in the discussion setion and Appendix D). Here we list some of them:
> >   1. *Consistency among multiple responses is more effective compared to verbalized confidence ($M=1$)*.
> >   2. *Aggregation based on answers and verbalized confidences outperforms aggregation based on answers only*.
> >   3. *Balancing efficiency, simplicity, and effectiveness through multiple benchmark datasets, we recommend the practitioners use “Top-K prompt + Self-Random sampling + Avg-Conf/Pair-Rank aggregation”.*
> >
> > - **Comparison with white-box models**:
> >     - While our method is centered on black-box settings, comparing it with white-box methods helps us understand the progress in the field. We newly conducted comparisons on five datasets with three white-box methods and observed that although white-box methods indeed perform better, the gap is narrow, e.g., 0.522 to 0.605 in AUROC. This finding underscores that the field remains challenging and unresolved.
> >
> > - **Novelty**: As pointed out by multiple reviewers, our paper makes several original contributions:
> >     - **As a strong baseline**: We have conducted an extensive benchmark analysis of black-box methods for confidence elicitation, analyzing the pros and cons of each strategy, setting strong baselines for future research. Such a study had never been done before and was arguably needed.
> >     - **As a comprehensive evaluation framework**: We have established an evaluation framework that emphasizes the importance of considering multiple evaluation metrics. We believe our framework should be used by the LLM research community as a standard pipeline to measure their LLMs' ability to express uncertainty.
> >     - **As an empirical evaluation**: We believe that empirical evaluations such as ours play an important role in providing a fair and comprehensive comparison of existing approaches to improve our understanding of a topic.
> >     - **Practical insights**: as mentioned by reviewer uQRx, we provide many insights that are not available from the traditional line of research, such as in calibrating image classification models:
> >         - We observe that there is a growing trend in using verbalized confidence for downstream tasks. Our research aims to highlight the overconfidence of LLMs, and its potential risks, e.g., the good performance might not be transferred to another dataset, for which a comprehensive evaluation is needed.
> >         - We analyze which strategies are effective and discuss the reasons behind their success or failure, serving as a foundation for future research to build upon.

---

### Meta-Review · Area_Chair_b7CC · 2023-12-07

**Metareview:**

This paper systematically evaluates the ability of LLMs to express uncertainty in their outputs. The focus is on "black box" systems. The authors consider various prompting strategies to elicit expression of uncertainty, as well as sampling and aggregation based methods. The work provides a number of interesting empirical findings that indicate LLMs appear to be "overconfident" in their outputs.

There was consensus that this is an important and timely problem. Despite what might be viewed as the limitation of focussing only blackbox models (precluding meaningful analysis of internals and how they relate to confidence), the findings here will likely be of interest to the community. The addition of "white-box" experiments (per reviewer uQRx's suggestion) is welcome, and I'd encourage mentioning these in the main body. While some reviewers expressed concerns regarding methodological novelty, the contribution here is fundamentally empirical, so this does not strike me as an issue.

**Justification For Why Not Higher Score:**

I think this paper will be of interest to the ICLR community, but it's somewhat limited scope (and the black-box setting along with limited datasets in particular) somewhat hinder its likely impact.

**Justification For Why Not Lower Score:**

The analysis offers interesting (if not entirely unpredictable) findings on a timely topic.

---

### Decision · Program_Chairs · 2024-01-16

Accept (poster)